



# How does the phytoplankton-light feedback affect marine N2O inventory?

Sarah Berthet [1*], Julien Jouanno [2], Roland Séférian [1], Marion Gehlen [3], William Llovel [4]

[1] CNRM, Université de Toulouse, Météo-France, CNRS, Toulouse, France
[2] LEGOS, Université de Toulouse, IRD, CNRS, CNES, UPS, Toulouse, France
[3] LSCE, Institut Pierre Simon Laplace, Gif-Sur-Yvette, France
[4] LOPS, CNRS/University of Brest/IFREMER/IRD, Brest, France

(*correspondence: sarah.berthet@meteo.fr)

## Abstract

The phytoplankton-light feedback (PLF) depicts how phytoplankton biomass interacts with the downwelling shortwave radiation entering the ocean. Considering the PLF allows differential heating across the ocean water column as a function of the phytoplankton concentration. Only one third of the CMIP6 Earth system models include a complete representation of the PLF. In other models, the PLF is mimicked either thanks to a prescribed climatology of chlorophyll or not represented at all. Consequences of an incomplete representation of the PLF on the marine biogeochemical content haven't been assessed yet and remain a source of multi-model uncertainty in future projection. Here, we scrutinize with a single modelling set-up how various representation of the PLF can impact ocean physics and ultimately marine production of a major greenhouse gas, the nitrous oxide (N2O). Global sensitivity experiments considering the ocean, sea ice and marine biogeochemistry have been performed at 1-degree of horizontal resolution over the last two decades (1999-2018). We show that the representation of the PLF has significant consequences on the ocean heat uptake and temperature of the first 300 meters of the tropical ocean. Temperature anomalies due to an incomplete PLF representation drive perturbations of the ocean stratification, dynamics and oxygen concentration. Different projection pathways for N2O production result from the choice of the PLF representation. Considering an incomplete representation of the PLF overestimates the oxygen concentration in the North-Pacific oxygen minimum zone and underestimates the local N2O production. This leads to important regional differences of sea-to-air N2O fluxes: fluxes are enhanced by up to 24% in the south Pacific and south Atlantic subtropical gyres, but reduced by up to 12% in oxygen minimum zones of the northern hemisphere. Our results based on a global ocean-biogeochemical model at CMIP6 state-of-the-art thus shine a light on a current uncertainty of the modelled marine nitrous oxide budget in that climate models.

## Plain language summary

Phytoplankton absorbs the solar radiation entering the ocean surface, and contributes to keep the associated energy in surface waters. This natural effect is not commonly represented in the oceanic part of climate models, or often suffers simplifications. We show that an incomplete representation of this biophysical interaction affects the way climate models capture ocean warming, what in turn uncertains the forecast of oceanic emissions of an important greenhouse gas called the nitrous oxide.

**Key-words:** phytoplankton-light interaction; bio-physical feedback; nitrous oxide; N2O; CMIP6 Earth system models; CNRM-ESM2-1; ocean-biogeochemical model; greenhouse gazes; marine emission; climate



**Key points:**

- forced ocean-biogeochemical experiments reveal that marine production of nitrous oxide is sensitive to the representation of the phytoplankton-light feedback
- the phytoplankton-light feedback perturbs the accumulation of heat and the ocean dynamics which drive changes in nitrous oxide production patterns
- an incomplete phytoplankton-light feedback overestimates sea-to-air N2O fluxes by up to 24% in subtropical gyres and reduces them by up to 12% in oxygen minimum zones

## 1. Introduction

Couplings between the physical, biogeochemical, or ecosystem compartments of the ocean can induce abrupt system changes (Heinze et al., 2021). Currently, the only coupling in Earth system models existing between modelled marine biogeochemistry and ocean dynamics is the interactive phytoplankton-light feedback (PLF) (Séférian et al., 2020). It is at play when the chlorophyll (CHL) produced by the biogeochemical compartment is used to determine the fraction of shortwave radiation (SW) penetrating into the ocean surface waters. In this case, the CHL concentration mimicking the influence of the marine biota on the vertical redistribution of heat in the upper ocean is consistent (because the same) with the one used to compute biogeochemical cyclings.

### a) Phytoplankton-light feedback (PLF)

Since the first observational evidences on how surface materials may impact light absorption by the ocean and change the radiative imbalance within the mixed-layer depth (Kahru et al. 1993), several ocean models have gradually accounted for this biophysical interaction. Gildor and Naik (2005) highlighted the importance to consider monthly variations of CHL to capture the first-order effect of marine biota on light penetration in ocean models. Then, introducing a light-interactive CHL in numerical experiments has been shown to affect oceanic phenomenons on a wide range of spatial and temporal scales. Enabling a phytoplankton-light interaction modifies the intensity of the spring-bloom in subpolar regions (Oschlies, 2004), the maintenance of the Pacific Cold Tongue (Anderson et al., 2007), the seasonality of the Arctic Ocean (Lengaigne et al., 2009), the strength of the tropical Pacific annual cycle and the ENSO variability (Timmermann and Jin, 2002; Marzeion et al., 2005), the northward extension of the meridional overturning circulation (Patara et al., 2012), and the cooling of the Atlantic and Peru-Chili upwelling systems (Hernandez et al., 2017, Echevin et al., 2022).

However the mean effect of the PLF on sea surface temperature (SST) has been argued to depend on the numerical framework (forced ocean versus coupled ocean-atmosphere models). The conflicting results obtained on that topic in the literature have been shown to be mainly due to diverging bio-optical protocols among models rather than by the inclusion of air-sea coupling. Following Park et al. (2014) the atmosphere-ocean coupling just acts to amplify the PLF-induced mean change, but does not alter the sign of the response obtained in ocean-only experiments. Two main causal chains have been proposed to interpret the sign of the final heat perturbation, opposing the proeminence of an indirect dynamical response (Murtugudde et al., 2002; Löptien et al., 2009) to that of a direct thermal effect (Mignot et al., 2013; Hernandez et al., 2017). Regional dependencies of these two mechanisms have also been evidenced (Park et



al., 2014). However, beyond the diversity of model responses, a consensus emerges about the
first order effect the PLF exerts on the ocean, being to perturb the ocean thermal structure
(Nakamoto et al., 2001; Murtuggude et al., 2002; Oschlies, 2004; Manizza et al., 2005, 2008;
Anderson et al., 2007; Lengaigne et al., 2007; Gnanadesikan and Anderson, 2009; Löptien et
al., 2009; Patara et al., 2012; Mignot et al., 2013; Hernandez et al., 2017). By trapping more
heat at the ocean surface in eutrophic regions, such as coastal or equatorial upwellings areas,
the presence of phytoplankton increases the surface warming. Confining heat at the surface
leads to less heat penetrating into the ocean interior. Because these effects depend on upper
ocean stratification, they are more active during local summer and at low latitudes. An
important role is attributed to modelled seasonal deepening of the mixed layer as it determines
the intensity of the underlying temperature anomaly and its vertical movement to the surface.
In other terms, whatever the temporality of the causal chain, changes in the PLF representation
are expected to both pertub the ocean heat uptake, and trigger perturbations of both the water
column stratification and associated ocean dynamics.
**b) This study: implications for N2O budget uncertainties**
Nitrous oxide (N2O) is a major ozone-depleting substance (Ravishankara et al., 2009; Freing et
al., 2012) and a potent greenhouse gas, whose global warming potential is 265-298 times that
of CO2 for a 100-year timescale (Myhre et al., 2013). The spatial coincidence between marine
productive areas and observed hot-spots of N2O production leads to question the impact of an
incomplete representation of the PLF on the simulated N2O inventory. Indeed recent
observational studies highlight that N2O production is high in low-oxygen tropical regions and
cold upwelling waters (Arévalo-Martinez et al 2018; 2020; Yang et al., 2020; Wilson et al., 2020).
N2O has been shown to become more highly saturated in the surface waters of equatorial
upwelling regions due to the upward advection of N2O-rich waters (Arévalo-Martínez et al.,
2017). Thus, regions known to account for the most productive areas of the ocean spatially
coincide with the highest N2O production: 64% of the annual N2O flux occurs in the tropics,
and 20% in coastal upwelling systems that occupy less than 3% of the ocean area (Yang et al.,
125   2020).
Despite these recent results, a large range of uncertainties still surrounds oceanic N2O
emissions, as large areas of both the open and coastal ocean remain undersampled by
observations (Wilson et al., 2020). In particular, the sparcity of observational data in regions
which emit considerable amounts of this gas contributes to increase the uncertainties. The
recent global budget of Tian et al. (2020) estimates natural sources from soils and oceans
contributing up to 57% to the total N2O emissions for the recent decade, with the ocean flux
reaching 3.4 (2.5–4.3) Tg N yr$^{-1}$. But this oceanic contribution reflects a large uncertainty range,
as it has been computed based on global ocean-biogeochemical models (and moreover based
on a very small number of models). Indeed very few climate models, even in the current CMIP6
generation, include emissions (and beforehand a complete representation of N cycling) of N2O
fluxes: only 4 out of the 26 Earth system models considered in Séférian et al. (2020) model
marine N2O emissions.
Finally, this last generation of Earth system models projects an enhanced ocean warming in
response to climate change, which is in turn expected to increase upper-ocean stratification
(Sallée et al., 2021) and to contribute to greater reductions in upper-ocean nitrate and



subsurface oxygen ventilation (Kwiatkowski et al., 2020). Ocean warming and deoxygenation
constitute two triggers of high-probability high-impact climate tipping points (Heinze et al.,
2021), which have been identified as two of the main environmental factors influencing marine
nitrous oxide (N2O) distributions (IPCC, 2019; Hutchins and Capone, 2022). Through the
addition of its expected imprints on the upper ocean stratification, the PLF representation could
further change the oceanic N2O source by modulating the mixing between N2O-rich water and
intermediate depths, perturbing the way N2O-rich water reach the air-sea interface (Freing et
al., 2012).
In that perspective, the present study investigates how an incomplete representation of the
phytoplankton-light feedback may particularly uncertain nitrous oxide prediction in an up-to-
date global ocean-biogeochemical model making up the current generation of Earth system
models. Section 2 describes the numerical model and the set of experiments encompassing the
existing options to consider CHL modulations of the incoming SW radiation. The N2O
parametrization used in this model is also detailed. Section 3 presents the important effect of
an interactive PLF on the ocean heat content, associated ocean stratification and dynamics,
and its repercussions on marine N2O inventory. Finally, Section 4 summarizes the main results,
addresses their broader implications, and discusses the future work motivated by this study.
## 2. Methodology
### a) Sensitivity experiments with a global ocean-biogeochemical model
Recent projections of future N2O emissions realized as part of intercomparison projects like
CMIP6 are still based on Earth system models with a low spatial resolution (Séférian et al.,
2020). For sake of coherence with CMIP biogeochemical modelling efforts, we use a global
ocean-biogeochemical configuration of the NEMO-PISCES model (Madec, 2008; Aumont et al.,
2015) at 1° of horizontal resolution in the following. This model corresponds to the oceanic
component of CNRM-ESM2-1 (Séférian et al., 2019) and is one of the few CMIP6-class models
that contributed to the Global N2O budget (Tian et al., 2020). Details on model configuration
are given in Berthet et al. (2019). Using an ocean-only configuration allows to isolate the local
response induced by the PLF, by not confounding it with potential inter-basin feedbacks acting
through the atmosphere.
The global ORCA1 domain was first spun-up under preindustrial conditions during 2000 years
by cycling the first 5 years of the JRA55-do atmospheric reanalysis (Tsujino et al., 2018; i.e.
OMIP2 compliant: Tsujino et al., 2020). The cycling method was prolonged until year 1958,
while considering the historical evolution of atmospheric CO2 and N2O since year 1850. Then
the complete period of JRA55-do atmospheric forcing has been rolled out from 1958 to 2018.
This first experiment (hereafter chl_inter) together with the spin-up both account for an
interactive PLF: the penetration of SW radiation into the ocean surface is constrained by the
CHL concentration produced by the PISCES biogeochemical component. Hereafter the term
"spin-up" has been extended to the whole period simulated before year 1999.



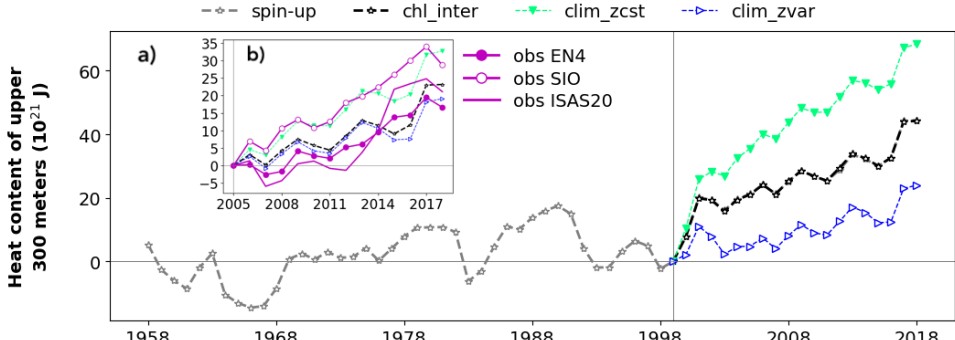

Figure 1: Modelled **tropical [35°S-35°N] heat content of upper 300 m** (OHC300; in ZJ) for each
experiment described in Table 1: chl_inter (black; empty stars), clim_zcst (green; full downward
triangles) and clim_zvar (blue; empty rightward triangles). In (a) final part of the spin-up has
been added in gray to illustrate the branching protocol in year 1999, and OHC300 anomalies
have been computed with respect to year 1999. Subplot (b) zooms over the Argo period to
compare modelled tropical OHC300 anomalies with 3 in situ-based products (see section 2c).
At year 1999 two other sensitivity experiments were branched off (Figure 1). The experiments
clim_zcst and clim_zvar account for an incomplete and external PLF, as they consider an
observed climatology of surface CHL from ESACCI (Valente et al., 2016) in order to compute the
light penetration into sea water. These two experiments differ from each other by the "realism"
of the vertical profile derived from the ESACCI CHL surface climatology (Table 1). The vertical
profile used for clim_zcst is considered costant and spreads uniformly in the vertical direction
to the level below which we consider the light cannot penetrate (Figure S1, b and d-f). The CHL
vertical profile of clim_zvar is variable and derived from the 2D surface climatology following
Morel and Berthon (1989) to the level of light extinction (Figure S1, c and d-f). This set of
experiments is representative of the several configurations used in the case of CMIP
intercomparison project.

| Experiment | Which CHL fields to interact with incoming SW radiation? | PLF nature |
|---|---|---|
| chl_inter | uses directly the 3D CHL produced by the biogeochemical component | interactive |
| clim_zcst | uses the prescribed monthly climatology of ESACCI CHL with a constant vertical profile, equal to the value of the surface climatology up to the level of light extinction | incomplete |
| clim_zvar | uses the prescribed monthly climatology of ESACCI CHL with a variable vertical profile, derived from the surface climatology following Morel and Berthon (1989) | incomplete |

Table 1: Experimental set-up.



Note that in clim_zcst and clim_zvar, CHL concentrations used for radiation and
biogeochemical cycles are decoupled: the biogeochemical model produces CHL and uses it for
biogeochemical element cycling but feedback of CHL on physics (stratification, ocean heat
content) is determined by the externally prescribed CHL climatology. In this case the marine
biota computed by the biogeochemical model does not affect the physical properties of the
ocean waters.
Finally, all three experiments use a simplified formulation of light absorption by the ocean to
calculate both the phytoplankton light limitation in PISCES and the oceanic heating rate
(Lengaigne et al., 2007). In this formulation, visible light is split into three wavebands: blue
(400–500 nm), green (500–600 nm) and red (600–700 nm); for each waveband, the CHL-
dependent attenuation coefficient is fitted to the coefficients computed from the full spectral
model of Morel (1988) (as modified by Morel and Maritorena (2001)) assuming the same
power-law expression.
Consequences on the marine biogeochemical mean state of incomplete representations of the
PLF are assessed in the following by difference with our control run chl_inter. This methodology
allows to evaluate how different levels of realisms and complexity in resolving bio-physical
interactions impact the physical and biogeochemical content of the modelled ocean.
**b) N2O parameterization**
As described by Aumont et al. (2015), PISCES models five limiting nutrients for phytoplankton
growth: nitrate and ammonium, phosphate, silicate and iron. The phosphate, nitrate-
ammonium nutrient pools are not really independent in PISCES, as they are linked by a constant
and identical Redfield ratio in all the modelled organic compartments. Redfield ratios are set to
122:16:1 for C:N:P following Takahashi et al. (1985) and the -O:C ratio is set to 1.34 (Kortzinger
et al., 2001).
In the ocean, N2O production and consumption are driven by marine bacteria in slightly
oxygenated waters. N2O can occur as a by-product during microbial nitrification and as an
intermediate product during denitrification (Freing et al., 2012). The oxic-anoxic interface
above oxygen minimum zones (OMZ) has been shown to provide appropriate conditions to
enable N2O production (Ji et al., 2018). In the absence of oxygen, nitrate ($NO_3^-$) is the next
preferred electron acceptor for respiration after oxygen according to the electrochemical series
(Lam and Kuypers, 2011). While nitrification is typically assumed to be an aerobic process,
substantial suboxic nitrification has also been reported in many of the world ocean's major
suboxic zones. Denitrification has been shown to be the dominant process for N2O production
in the southern (Ji et al., 2015, 2018) and northern (Ji et al., 2018) part of the Pacific OMZ, but
uncertainty still subsist: Kalvelage et al. (2013) observe that water-column denitrification was
only of minor importance (<<1% total N loss) for the overall N budget in the eastern tropical
South Pacific OMZ and that anammox was the dominant mode of N loss at the time of sampling.
The bacterial pool is not yet explicitly modelled in PISCES. Processes of N2O production like
nitrification or denitrification are not formally expressed, and PISCES diagnoses their effects
from specific environmental conditions. Such modelling approach with an indirect
representation of the N2O yield is rather common in present Earth system models due to the





complexity of involved processes (Battaglia and Joos, 2018). For example, in MPI-ESM 1-2-LR
(Ilyina et al., 2013) and MIROC-ES2L (Hajima et al., 2020), two of the few other Earth system
models simulating marine N2O emissions in CMIP6 (Seferian et al., 2020), the production of
N2O is mainly linked to the consumption of oxygen (O2) during remineralization of organic
matter.
In PISCES it is assumed that the distribution of nitrifying bacteria in the model is ubiquitous in
the ocean interior, so wherever there is export of organic matter to depth the model computes
nitrification, consuming ammonium and producing nitrate (Martinez-Rey et al., 2015).
Nitrification is particularly enhanced in total absence of light, whereas oxygen levels should be
above the suboxic threshold of 1 μmol L$^{-1}$. Denitrification is computed in the model where
dissolved oxygen concentration falls below 5 μmol L$^{-1}$, which defines suboxic waters (Cocco et
al., 2013; Bopp et al., 2013).
In each grid point below 100 m depth (as N2O production is inhibited by light), a unitless
function f(O$_2$) depending on the oxygen concentration [O$_2$] (in μmol L$^{-1}$) is computed following:
$f([O_2] < 1 \; \mu mol \; L^{-1}) = [O_2]$
$f(1 \; \mu mol \; L^{-1} \leq [O_2] \leq 5 \; \mu mol \; L^{-1}) = 1$          (1)
$f([O_2] > 5 \; \mu mol \; L^{-1}) = 0.7*exp(-0.1*([O_2] - 5)) + 0.3*exp(-0.01*([O_2] - 5))$
f(O$_2$) allows to evaluate the suboxic production of N2O based on Martinez-Rey et al. (2015):
$[N2O]_{suboxic} = \alpha + \beta * f(O_2)$                                    (2)
with $\alpha$ being the nitrification coefficient for N2O background production equal to 10$^{-4}$ molN2O
per molO$_2$ consumed. $\beta$ is the denitrification coefficient which scales the oxygen-dependent
function. It is equal to 30 10$^{-4}$ molN2O per molO$_2$ consumed.
Then the local trend of nitrous oxide concentration [N2O] is finally evaluated by Eq. (3) at each
time step as:
$d[N2O]/dt = \quad [N2O]_{suboxic} * zolimit * o2ut$              (3.1)  remineralization
$\qquad\qquad\qquad - sink_{N2O} * [N2O]$                          (3.2)  sink term
$\qquad\qquad\qquad + [N2O]_{suboxic} * zonitr * o2nit$           (3.3)  nitrification
$\qquad\qquad\qquad + [N2O]_{suboxic} * zgrazing * o2ut$          (3.4)  grazing
where in the first term (3.1) zolimit accounts for ammonification in oxic waters through oxygen
consumption during the remineralization of the organic matter at the o2ut ratio of 133/122. In
the second term (3.2) sink$_{N2O}$ is the N2O sink term coefficient corresponding to the N2O
consumed under anoxic conditions by denitrification at a rate of 7.12 10$^{-4}$ s$^{-1}$. The third term
(3.3) represents the part of N2O concentration produced as an intermediate product of
nitrification at a o2nit ratio of 32/122. The last term (3.4) produces N2O by grazing of the
remnant organic matter.
The N2O partial pressure difference across the air-sea interface (sea-to-air Dpn2o; in atm) is
then computed based on





306       $Dpn2o = [N2O]_{surface} / solub_{N2O} - pn2o * P_{atm}$       (4)
with pn2o, the atmospheric partial pressure of N2O equal to 273.021 ppb, $P_{atm}$ the atmospheric
pressure in $N/m^2$, and $solub_{N2O}$ the N2O solubility in $mol/m^3$ which depends on in-situ
temperature and practical salinity following the formulation of Weiss and Price (1980).
Finally sea-to-air N2O fluxes $(mol/m^2/s)$ are inferred based on Wanninkhof (1992; 2014):
314       $N2O\_flux = Dpn2o * solub_{N2O} * Kg_{N2O}$       (5)
with $Kg_{N2O}$ being the piston velocity for N2O (m/s), which depends on wind speed, ice fraction
and temperature .
**c) Observations**
Model results are compared with available observational-based gridded T/S datasets. Ocean
heat content (OHC) of the upper 0-300-meters layer has been inferred from three different
products: i) the global objective analysis of subsurface temperature EN4 (Good et al., 2013), ii)
the SIO product of the Scripps Institution of Oceanography (Roemmich anf Gilson, 2009), and
iii) the ISAS20 optimal interpolation product released by the Ifremer (Kolodziejczyk et al., 2019;
Kolodziejczyk et al., 2021). While SIO and ISAS20 products consider only Argo T/S profiles, EN4
dataset considers all types of in situ profiles providing temperature and salinity (when
available). These three in situ-based datasets are considered since 2005, when the Argo
coverage became sufficient to characterize the global ocean. Details on OHC computation can
be found in Llovel and Terray (2016) and Llovel et al. (2022). The text also refers to cross-
validations performed on OHC of the deeper layers (0-700 m and 0-2000 m) that have been
performed with OHC anomalies from World Ocean Atlas 2009 (Levitus et al., 2012). A monthly
climatology (1955-2012) of oceanic temperature from World Ocean Atlas 2013 version 2
(Locarnini et al., 2013) has been used to evaluate modelled temperatures. Modelled O2 has
been compared with the annual climatology of O2 from World Ocean Atlas 2013 (Garcia et al.,
2014). The recent dataset of Dpn2o observations compiled by Yang et al. (2020) is used to
evaluate modelled Dpn2o.
**3. Results**
**a) Impact of PLF on the upper ocean heat content and dynamics**
Meridional sections reveal that heat perturbations in response to changing CHL fields
interacting with light are limited to the top 0-300 m layer of the ocean and predominantly affect
the tropical area (*Figure 2* and Figure S2, c-d).




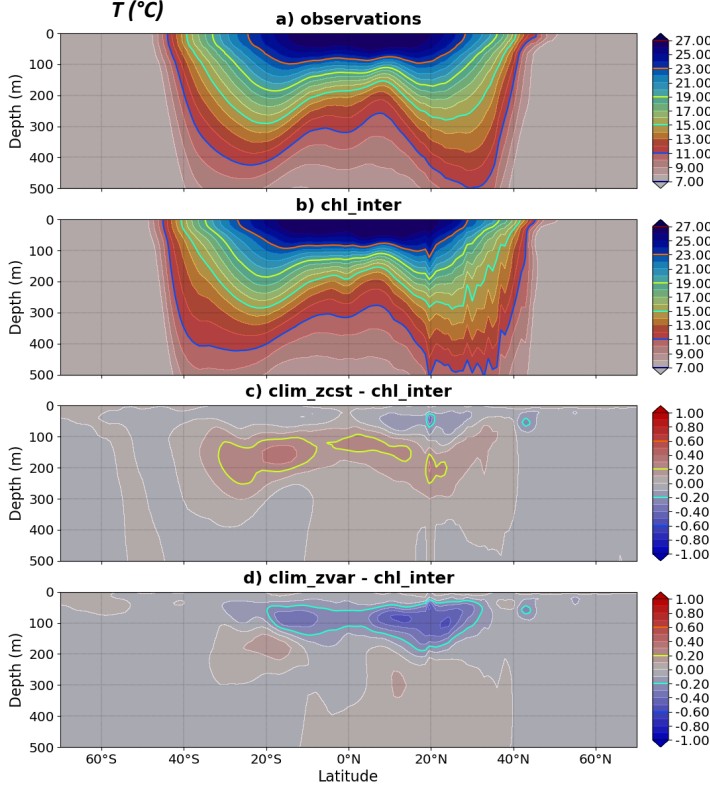

Figure 2: Mean 2009-2018 meridional section of **temperature** (°C) averaged over the whole tropical band (0-360°E) for a) observations, b) chl_inter and its differences with c) clim_zcst and d) clim_zvar.

The largest temperature anomalies are observed near the thermocline depth and reflect upper
ocean warming and deepening of the thermocline in clim_zcst (*Figure 2*c), and cooling and
shallowing of the thermocline in clim_zvar (*Figure 2*d). In clim_zcst the ocean warming reflects
large-scale patterns of a tropical CHL deficit compared to chl_inter (Figure S1, b). Temperature
differences seen in the near-surface layer (0-50 m) is lower than that of the 50-300 m layer.
This is expected as a result from weak stratification but also from experiments run with a forced
atmosphere in which the temperature of the ocean surface layer is constrained by the
atmospheric prescribed state.
When using an incomplete representation of the PLF, two contrasting trends of the upper
ocean heat content (OHC) emerge compared to our control run chl_inter (Figure 1a).
Over the Argo period (2005-present) EN4 estimates of tropical OHC300 are in very good
agreement with our warmest experiment clim_zcst (Figure 1b), while the two other
dataproducts SIO and ISAS20 are in better agreement with our control run chl_inter and with
clim_zvar. Note that the good accordance between modeled OHC300 and observations is not



to take for granted (Cheng et al., 2016; Liao et al., 2022) and that non-negligible differences
among OHC dataproducts exist and are generally particularly strong in the upper 0-300 m layer
(Lyman et al., 2010; Liang et al., 2021). The spread between these products at the end of the
2005-2018 period (12.1 $10^{21}$ J) is comparable with that of our numerical set (13.6 $10^{21}$ J). The
modelled OHC in chl_inter is in very good agreement with current global mean in situ
observations (Meyssignac et al., 2019; see their Figure 11) and with OHC anomalies derived
from WOA09 (Levitus et al., 2012). In accordance with these observations, our ocean-
biogeochemical model simulates a global mean increase of OHC over the 2006-2016 period of
order 40 $10^{21}$ J for the upper 700 m, and of about 70 $10^{21}$ J for the 0-2000 m layer.
Subsurface thermal anomalies develop rapidly (Figure S3) after branching of clim_zvar and
clim_zcst in 1999. The dipole structure of the anomaly seen in clim_zcst reflects the surface
heat trapping in chl_inter and the associated subsurface cooling (Figure S3, b). Indeed in
clim_zcst the vertically constant and weaker profiles of CHL trap less incoming SW than the CHL
maximum seen in chl_inter between 0 and 100 m depth (Figure S1, d-f). The negative anomaly
in clim_zvar suggests that the parameterization of Morel and Berthon (1989) contributes to
underestimate the ocean heat uptake (Figure S3, c and Figure S2, d) by comparison to chl_inter.
This heat deficit results from the overestimation of the vertical integral of CHL over large areas
of the tropical domain in clim_zvar compared to chl_inter (Figure S1, c), which catch the energy
associated to the incoming radiation without distributing it to the water column.
In both clim_zcst and clim_zvar the subsurface temperature anomaly deepens progressively
over the first six years of simulation as a result of vertical mixing (Figure S3). This evolution
indicates that part of the OHC300 differences between experiments comes from the
adjustment of clim_zcst and clim_zvar to the spin-up mean state yielded by an interactive PLF.
However, differences in OHC300 from experiments having spin-ups consistent with their own
PLF representations are expected to be even greater. The range of uncertainties evaluated here
should be considered at the lower end of estimate of OHC discrepancies that may emerge from
changing the PLF representation.
Prescribing a constant vertical profile of CHL (clim_zcst; green) to compute the penetration of
the radiation into the ocean increases the OHC of the upper 300 meters (hereafter OHC300)
by more than 20 $10^{21}$ J during the last two decades (1999-2018) compared to chl_inter (Figure
1). This rise of OHC300 decreases the vertically-weighted sum of the tropical potential density
of the upper 300 m at the end of the simulated period by 5 kg/m$^3$ compared to chl_inter (Figure
S4). Surprisingly, the opposite trend (a reduced OHC300 compared to chl_inter) is simulated
with the same state-of-the-art CMIP6 ocean-biogeochemical model when considering a
variable vertical profile of CHL (clim_zvar; blue). However Figure 1 highlights that the simulation
using a consistent CHL for interacting with both incoming SW and biogeochemical cyclings
(chl_inter) does not amplify one of these two trends, as clim_zcst and clim_zvar surround
chl_inter. Average ranges of uncertainties over the extended tropical domain (35°S-35°N)
exceed 40 $10^{21}$ J in terms of OHC300 (Figure 1), 4 meters for the thermocline depth and more
than 9 kg/m$^3$ for the potential density perturbation (Figure S4).
Similar to OHC300, ranges of uncertainty for the OHC estimates of deeper layers (0-700 m and
0-2000 m) also slightly exceed 40 $10^{21}$ J. Such uncertainty ranges are quite important as they
have been obtained by only changing the PLF representation in a single ocean-biogeochemical





model. By comparison, in the context of OMIP protocols, Tsujino et al. (2020) give spreads
between CMIP models estimates of order 50 $10^{21}$ J for the OHC of the upper 700m after 20
years (see their Figure 24, a-b). Regarding the OHC integrated over the 0-2000m layer, they
present an inter-model spread between 50 and 100 $10^{21}$ J, depending on the OMIP protocol
considered (see their Figure 24, d-e). So, the OHC300 uncertainty of 40 $10^{21}$ J triggered by the
representation of the PLF in our set of experiments has a comparable order of magnitude than
the current multi-models estimation of OHC. Part of the OHC multi-model uncertainty in
current climate models may be due to different representations of the phytoplankton-light
interaction.
The heat and associated density perturbations also cause dynamical modifications of upper
ocean currents (Figure *3*). Absolute differences in upper ocean velocities (average between 0
and 300m depth) are between |0.05| and |0.6| cm/s with strongest differences  along the
equator revealing perturbations of the equatorial undercurrent (Figure *3*, b and c). Circulation
around the subtropical gyres is also impacted, in particular for the south-Pacific subtropical
gyre. These modifications of zonal and meridional dynamics spread over the entire tropical
latitudes, from 30°S to 30°N, strongly supporting the idea that heat perturbations induced by
modifying interactions between CHL and incoming SW cause non-negligible modifications of
the equatorial and tropical ocean dynamics.

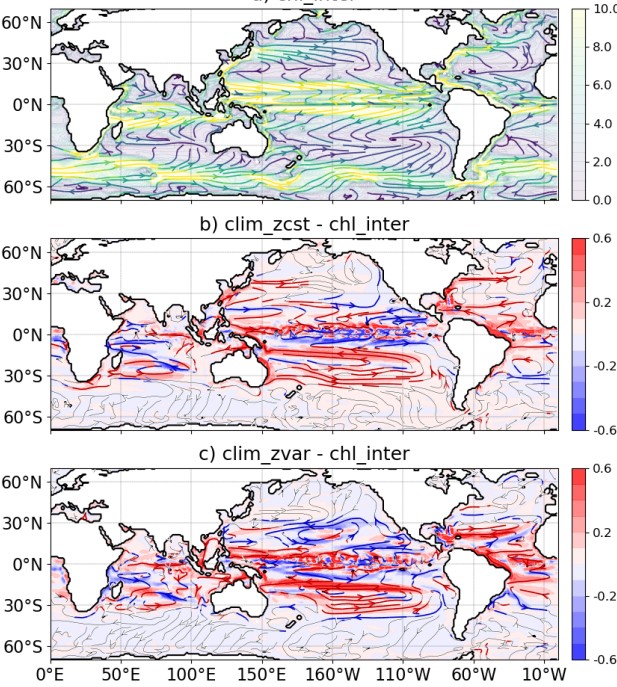

Figure 3: Annual mean speed (color; cm/s) and streamlines of **oceanic currents** between 0-300
m over the 2009-2018 period for a) chl_inter, and its differences with b) clim_zcst and c)
clim_zvar. In b-c) streamlines are colored when absolute speed are larger than 0.05 cm/s.



**b) PLF impact on N2O production**
Perturbations of the annual pycnocline depth (Figure 4, a-c) highlight a vertical adjustment to
the large-scale dynamical anomalies (Figure 3). Variations of the pycnocline integrate
perturbations of both thermal and salinity stratifications. However, in our experiments heat
anomalies appear to drive perturbations, and pycnocline depth anomalies mainly reflect those
of the thermocline. The cold anomaly dominating the tropical domain in clim_zvar (Figure S2,
d) appears to be vertically redistributed, as it triggers the raising of the isopycnals (Figure 4, c).
In contrast to the anomalies seen over most of the tropical Pacific, a deepening of the
isopycnals reaching up to 20 meters is modelled in both South Pacific and Atlantic subtropical
gyres in clim_zcst and clim_zvar (Figure 4, b and c). Over these subtropical gyres heat is
redistributed along the vertical as the subsurface warm anomaly dives which in turn causes a
deepening of the pycnocline (Figure 4, b and c). As stressed by Sweeney et al. (2005), small
changes in CHL concentration (Figure S1) may have important effects on the mixed layer depth
in these subtropical gyres due to low local wind speeds. Strong winds would drive the mixed
layer depth independently of the CHL changes, explaining why the pycnocline is barely
perturbed along the equator (Figure 4, b and c). In line with their results, our set of experiments
highlights that small CHL changes in low productivity regions trigger a vertical redistribution of
density anomalies affecting the stratification.

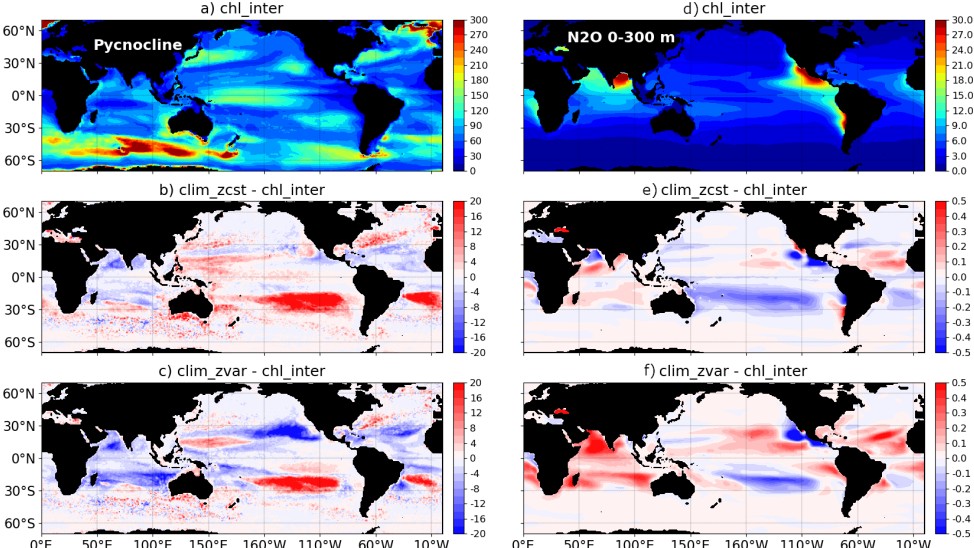

Figure 4: a-c) Depth of annual **pycnocline** (m) for 2009-2018, computed as the annual mean
depth of the maximum of the Brunt-Väisälä frequency $N^2$(T, S) over the water column (Maes
and O Kane, 2014). d-f) Mean **[N2O]** (micro-molN/m$^3$) over the first 300 meters depth for
(upper line) chl_inter and its mean-state differences with (middle) clim_zcst and (bottom)
clim_zvar.


Anomalies of N2O concentration integrated over the first 300 meters of the water column
(Figure 4, e and f) are in good agreement with patterns of pycnocline anomalies over the tropics
(Figure 4, b and c). These comparable spatial structures attest that N2O anomalies are driven
by perturbations of stratification in large parts of the tropical domain. Note that spatial patterns
are robust against expanding the column used to perform the mean of N2O concentration up
to 6000 meters, since most of the N2O perturbation is contained in the top 300 meters as
reported also for physical variables.
In the South Pacific subtropical gyre, the concomitance of i) an increased temperature (Figure
S2, c and d), ii) a reinforced transport  (Figure *3*, b and c) and iii) a weakened stratification
illustrated by a local deepening of the pycnocline (Figure 4, b and c), contributes to decrease
the N2O concentration in both clim_zcst and clim_zvar (Figure 4, e and f). By contrast, in the
South Indian Ocean and North tropical Atlantic the increase of N2O concentration seems to be
mainly driven by the mean shoaling of the local pycnocline, as both regions exhibit contrasted
perturbations in terms of transport and temperature. Finally, in the North-Pacific OMZ area,
the strong N2O deficits in both clim_zcst and clim_zvar do not respond to stratification and
transport anomalies but are rather driven by a local rise of O2 concentration (Figure S5).
Considering an incomplete PLF contributes to overestimate the oxygen concentration in this
OMZ and leads to a lack of local N2O production.
The relationship between N2O concentration and OHC300 in the Tropical Ocean is inferred
next based on the three 20-years simulations (Figure *5*). Approaching the slope of the simulated
distributions by a linear regression gives quite distinct tropical N2O production pathways along
time as a function of the oceanic heat uptake: from 0.3 micro-molN $m^{-2}$ per ZJ for the most
simplified PLF scenario clim_zcst, to 1 micro-molN $m^{-2}$ per ZJ for clim_zvar. The slope of the
experiment with the higher level of realism in terms of interactivity (chl_inter) appears a
solution between the two previous extremes, as it increases its N2O production by 0.8 micro-
molN $m^{-2}$ per ZJ. Each of these N2O production pathways will not forecast the same temporal
evolution of the N2O budget and hence, the same climate in future. This result stresses the
importance of having an interactive PLF in order to neither overestimate nor underestimate
the N2O production forecast due to simplified representation of the PLF.

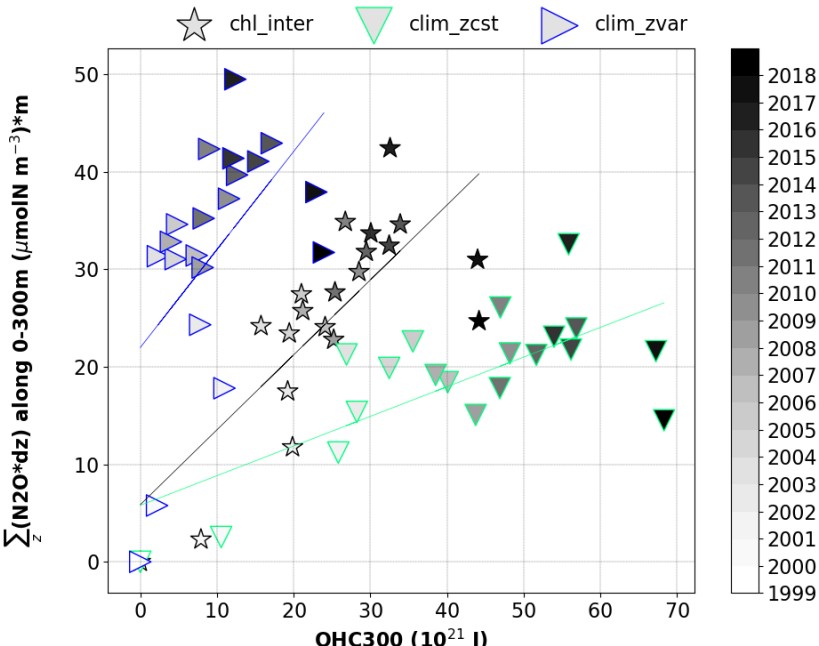


Figure 5: Annual N2O inventory over the first 300 meters depth (micro-molN/m$^2$/yr) as a
function of the annual OHC300 (ZJ/yr) and annually averaged over an extended tropical
domain (35°S-35°N).

**d) Repercussions on oceanic N2O emissions**

By perturbing the OHC, the ocean dynamics and the N2O production, the degree of realism of
the PLF has non-negligible consequences on Dpn2o and thus on N2O emissions at the air-sea
interface (Figure 6). Because the atmospheric partial pressure of N2O is identical among
experiments, differences in Dpn2o are driven by differences in surface N2O concentration
normalized by those in N2O solubility. It results that spatial patterns of Dpn2o anomalies (Figure
*6*) reflect differences in surface N2O concentration.

Interestingly, compared to a scenario considering a fully interactive PLF (chl_inter), an
incomplete representation of the PLF underestimates Dpn2o in all OMZ regions of the northern
hemisphere, which are strong emission zones (Figure *6*, c and d). Large Dpn2o anomalies of -
2.5 atm encompasses northern OMZ regions of the Indian, Pacific and Atlantic oceans and
anomalies reach up to -5 natm locally. Consequently, clim_zcst and clim_zvar underestimate
N2O fluxes by more than 12% in these OMZ regions compared to chl_inter. This result highlights
that the way to represent the PLF can be an important source of uncertainty in modelling N2O
fluxes. As a matter of fact, the oceanic contribution to the recent global N2O budget by Tian et
al. (2020) is based on only five global ocean-biogeochemical models (as still only few models
simulate marine N2O emissions). These models have different configurations of the PLF which
adds considerable uncertainty to simulated marine N2O emissions.


In subtropical gyres, the strong and direct effect of temperature (Figure S2, c and d) on [N2O]
(Figure 4, e and f) is in line with Yang et al. (2020) who show that a solubility regime drives the
seasonality of Dpn2o in that regions. Both clim_zcst and clim_zvar overestimate Dpn2o in
subtropical gyres of the South Pacific and South Atlantic (Figure *6*, c and d). This leads to an
overestimation of the regional N2O fluxes by 24% compared to a simulation having a complete
and interactive PLF representation (chl_inter).

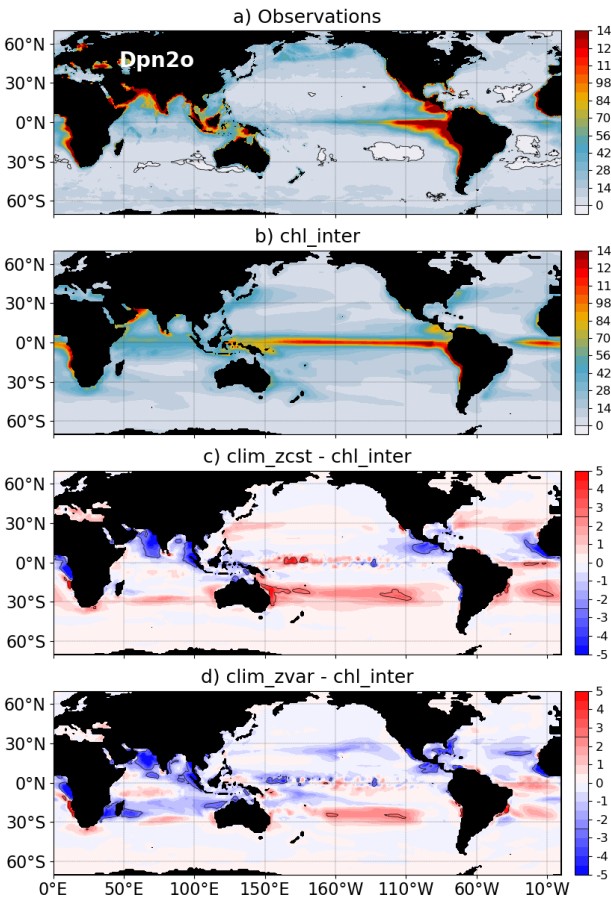

Figure 6: Mean sea-to-air **Dpn2o** (natm) computed from a) observations, b) chl_inter over the
2009-2018 period, and its differences with c) clim_zcst and d) clim_zvar compared to chl_inter.
**4. Conclusion**
In this study we use the ocean component (including ocean physics, sea ice and marine
biogeochemistry) of a global Earth system model that contributed to the last CMIP project
(CMIP6). Our ocean-biogeochemical model is one of the few currently able to represent an
interactive phytoplankton-light feedback (PLF) by constraining the penetration of shortwave
radiation into the ocean as a function of the chlorophyll concentration produced by the



biogeochemical model. Three experiments have been run at the horizontal resolution currently
used for intercomparisons of Earth system models (1°). Analyses are based on differences
between a control run with an interactive PLF (chl_inter) and two experiments using an
incomplete PLF (clim_zcst and clim_zvar) characterized by the use of a prescribed CHL
climatology to interact with the incoming solar radiation. Changing the way to compute how
the CHL filters the light penetration into the ocean reveals specific impacts of using an
interactive PLF.
Our results show that the strategy used to account for the impact of the biology on light
penetration significantly interfers with upper ocean heat uptake (Figure 1), and the associated
dynamics (Figure *3*) and stratification in the tropics (Figure 4, a-c). Our set of forced ocean-
biogeochemical experiments reveals that marine production of nitrous oxide (N2O) is sensitive
to the representation of the PLF (Figure 4, d-f). The heat perturbations add to the uncertainty
of modelled oceanic N2O production, and result in three N2O production trajectories with time
(Figure *5*) that in turn trigger regional differences of Dpn2o and sea-air N2O fluxes (Figure 6).
Compared to an ocean model using a fully interactive PLF (chl_inter), an incomplete PLF results
in an overestimation of N2O fluxes by up to 24% in the south Pacific and south Atlantic
subtropical gyres, and their reduction by up to 12% in OMZ of the northern hemisphere. Our
results based on a model at CMIP6 state-of-the-art emphasize an overlooked important source
of uncertainty in climate projections of marine N2O production and in current estimations of
the marine nitrous oxide budget.
In subtropical gyres of the southern Hemisphere which are regions of low productivity, small
CHL changes have a strong and direct effect on temperature (Figure S2, c and d), on transport
(Figure *3*, b and c) and on the local stratification (Figure 4, b and c). These concomittant effects
result in a local decrease of the N2O concentration in both experiments having a simplified PLF
representation (clim_zcst and clim_zvar).
Our results also question the reliability of current modelled estimates of the area and volume
of OMZ, as well as their trends in a future climate. The expansion rate of O2-depleted waters
still remains unclear and its controlling mechanisms are not yet fully understood (and
represented in today's models). Observations assessed that oceans have already lost around
2% of the global marine oxygen since 1960 (Schmidtko et al., 2017). The expansion of OMZ is
expected to result in an increase of the volume of water suitable for denitrification and to have
an impact on the production and decomposition of N2O (Freing et al., 2012). Our set of
experiments highlights that an incomplete representation of the PLF underestimates the
expansion of oxygen-depleted waters over the 20 years of simulation in comparison to
chl_inter. In clim_zcst and clim_zvar the global volume (0-1000 m) of hypoxic water with [O2]
under 50 mmol m$^{-3}$ is up to 2.3 10$^{14}$ m$^3$ lower in 2018 compared to that of the control run
chl_inter. Thus an incomplete representation of the PLF might lead to an underestimation by
1.2 % of the modelled tropical volume of low-oxygenated waters after 20 years.
Recent regional studies demonstrated that the interactive PLF strongly affects upwelling
systems of the south Pacific and Atlantic oceans (Hernandez et al., 2017; Echevin et al., 2021).
Coastal upwellings are known to be a place of high N2O production, with an annual N2O flux
totting up 20% of the global fluxes while these systems occupy less than 3% of the ocean area
(Yang et al., 2020). However, in our results main modelled perturbations are rather localized



over OMZ or subtropical gyres (Figure 4; Figure *6*). While the latter regional studies have been
performed using horizontal resolutions suitable to represent the complex dynamics of coastal
upwellings (from 10 km to about 28 km), it is well-established that climate models resolution
(~1-degree of horizontal resolution) does not allow to resolve these dynamics. The present
framework was designed to evaluate the sensitivity of CMIP models to the representation of
the PLF so why it used the spatial horizontal resolution of CMIP-like experiments. A step further
would be to evaluate how this sensitivity depends on the horizontal resolution by running
experiments at higher resolution with the same climate model. This would help to better
determine how the resolution of coastal upwelling systems may impact the modelled N2O
inventory through different PLF representations, as well as the associated modelled range of
uncertainty.
**Code availability**
Sources for NEMO and PISCES codes are available from https://forge.nemo-ocean.eu/nemo.
**Acknowledgements**
The OHC data were collected and made freely available by the International Argo Program and
the national programs that contribute to it (https://argo.ucsd.edu, https://www.ocean-
ops.org). The Argo Program is part of the Global Ocean Observing System. R. Séférian
acknowledges the European Union's Horizon 2020 research and innovation program under
grant agreement No. 101003536 (ESM2025 – Earth System Models for the Future).

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
