# Peer review of "How does the phytoplankton-light feedback affect the marine N2O inventory? 1 2 3 Sarah Berthet 1\*, Julien Jouanno2, Roland Séférian1, Marion Gehlen3, William Llovel4 4 5 1 CNRM, Université de Toulouse, Météo-France, CNRS,"

_Earth System Dynamics, 2022_

## Referee Comment (RC1)

**Berthet et al. – How does the phytoplankton-light feedback affect marine N2O inventory?**

**Summary**

The phytoplankton-light feedback (PLF) is the mechanism defining the absorption of short-wave radiation by phytoplankton in the ocean. This mechanism affects the distribution of light in the water column and thus the oceanic temperature. As a consequence, the oceanic heat content and the sea-air gas fluxes are affected, altering the whole climate system.

This paper presents simulations with a complete or incomplete representation of PLF and describes ocean physics and biogeochemistry accordingly. To answer this question, the authors use an Earth system model with oceanic, sea-ice and marine biogeochemical components only. What is novel and interesting is that (1) depending on the representation of PLF, the climate system can be highly modified, and (2) simulated N2O budget are highly uncertain. I believe that this study deserves to be published and provides implications for future climate models development. However, before accepting this manuscript, I think that there are some aspects that could be improved. Especially the description of the model setup and the description of the processes explaining the results. I may have misread some things but I hope that the following comments will help the authors to improve their manuscript.

**Major comments**

Section 1
The authors seem to say that a consensus exists on the effect of PLF on the thermal structure of the ocean (line 96-102). However, several studies report an increase in SST due to PLF (e.g. Oschlies, 2004; Anderson et al., 2007; Lengaigne et al., 2009) while others report a cooling of the surface of the ocean (e.g. Nakamoto et al., 2001; Manizza et al., 2005; Löptien et al., 2009; Paulsen et al., 2018). So far, no consensus exists but the cooling effect reported in these studies might be due to the non- or weak coupling between the oceanic and atmospheric components of the models (Tian et al., 2021).

Section 2.a
I understand that model configuration has already been described in Berthet et al. (2019) but it would be appropriate to give more explanation on the model configuration for the readers not familiar with Berthet et al. (2019). For instance, what is the vertical resolution of the oceanic grid? What is the depth of the first oceanic layer? Is phytoplankton simulated only in the first oceanic layer or can they go further down the water column? Additionally, I couldn't really understand the description of the simulations from the text. However, from Fig.1 it is clear that the authors run a spin-up for 2000 years and then run their simulations for 18 years only. In total, they run their model for 2018 years. Is the model in steady state? Maybe the authors should also state that their results are the average of the last 10 years of the simulations. In their simulations, do they prescribe the atmospheric CO2 and N2O concentrations or do they prescribe atmospheric CO2 and N2O emissions? Does the

ecosystem modelled consider only bulk phytoplankton or does it consider also e.g., cyanobacteria, diatoms? Is phytoplankton growth limited by light, nutrient and temperature? I think it would also be good to give the absorption coefficients used to parameterize PLF or directly give the equation of PLF.

Section 2.b
The paragraph line 240-252 would better fit in the introduction.

Section 3.a
I would suggest to put the chlorophyll maps in the results section rather than in appendix because it helps to understand to results. Especially, PLF affects the OHC via chlorophyll concentration so it might be easier to follow and easier to understand if the chlorophyll concentration maps are directly in the results section.
Furthermore, it seems that the authors use their two simulations with incomplete representation of PLF (chl_zcst and chl_zvar) as upper and lower limit of their uncertainties (line 403-405). I wonder why they think that? Do they consider that chl_zcst is a simple simulation giving the minimum OHC300 while they consider chl_zvar as a complex simulation (close to reality) giving the "real" OHC300? And that chl_inter is somehow fluctuating between these two simulations?

Section 3.b
As in Sweeney et al. (2005), the authors state that small changes in chlorophyll concentration drives important changes of the mixed layer depth in the subtropical regions (line 447-453). However, they do not explicitly explain what are the mechanisms behind these changes in MLD. Is it due to PLF? What is the link between chlorophyll and MLD? I think what the authors want to say is that changes in chlorophyll drive changes in temperature via PLF, which in turn drives changes in thermocline/pycnocline and thus in MLD.
The authors state that the decrease/increase in N2O concentration is driven by different mechanisms depending on the region studied. For instance, the decrease in N2O concentration in the South Pacific is due to increased temperature, enhanced circulation and deepening of the pycnocline. But in the North Atlantic the increase in N2O concentration is only due to the shoaling of the pycnocline. Additionally, in the North Pacific, the decrease in N2O concentration is due to the higher O2 concentration. Why do we have different mechanisms involved at different places of the world? For instance, why does O2 concentration is important for the North Pacific region but not for the North Atlantic or South Pacific regions?

Section 4
In conclusion, the authors detail some results about oxygen without showing them previously. I think they should talk about these results in the result section rather than showing them suddenly in the conclusion section.
Furthermore, the authors should also discuss the fact that their model setup does not consider an atmospheric component. Do they expect similar results if they use a coupled ocean-atmosphere model rather than using an atmospheric N2O and CO2 forcing? For instance, Asselot et al. (2022) show that PLF affects the climate system mainly via sea-air greenhouse gas fluxes. Adding an interactive atmospheric component could therefore lead to higher greenhouse gases in the atmosphere, increasing the atmospheric temperature. The higher

atmospheric temperature might increase the oceanic temperature and thus enhance the effect of PLF on N2O fluxes.

**Specific comments**

Line 17: Replace "thanks to" by "by".

Line 22: Replace "experiments" by "simulations".
This comment is valid for the entire manuscript.

Line 23: Replace "have been performed" by "are performed".

Line 34-35: Replace "shine a light on a current uncertainty of the modelled marine nitrous oxide budget in that climate models." by "shine light on current uncertainties of modelled marine nitrous oxide budget in that climate model."

Line 41: Replace "suffers" by "undergoes"

Line 43: Replace "uncertains the forecast" by "leads to uncertain forecast"

Line 92-94: The authors state that "Two main causal chains have been proposed to interpret the sign of the final heat perturbation" but they do not give these two causal chains. Furthermore "causal chains" could be directly replaced by "causes".

Line 132: I guess the authors are speaking about atmospheric emissions here.

Line 132: Which decade?

Line 136: Replace "model marine" by "simulate marine"

Line 157: Parameterization.

Line 202-203: Rephrase please

Line 213: Replace "biogeochemical element cycling" by "biogeochemical cycles".

Line 325: Replace "by the Ifremer" by "by Ifremer".

Line 332-337: The authors compare their modelled temperature and oxygen with several database. Is it shown somewhere? In figures or appendix?

Line 388-392: I cannot say what the authors mean with these two sentences.

Line 393: green on Figure 1?

Line 394: directly use OHC300 as it is previously used.

Line 400: blue on Figure 1?

Line 427: Replace "modifying" by "different".

Line 442: Replace "raising" by "shallowing".

Line 445-447: "Over these … the pycnocline" please rephrase.

Line 460: Replace "upper line" by "upper panel".

Line 466-469: Please rephrase.

Line 474: Replace "By contrast" by "In contrast".

Line 484: Remove "next".

Line 484-486: "Approaching … heat uptake" please rephrase.

Line 500: Should be section 3.c rather than section 3.d

Line 502: Replace "the degree of realism of the PLF" by "the way PLF is simulated/modelled".

Line 504-507: Please rephrase because it seems that you consider that Dpn2o anomalies reflect only differences in surface N2O concentration while Dpn2o anomalies can also reflect differences in N2O solubility.

Line 512-513: Are the units atm or natm?

Line 548: Remove "and".

Line 551: Replace "add" by "added".

Line 552: Remove ", and".

Line 552: Remove "with time".

Line 556: Replace "their" by "a".

Line 591: Replace "so why it used" by "that is why it uses"

Figure S4: I don't understand how densities (y-axis) can be negative. Is it a density anomaly represented?

Figure 3a: The colorbar can be improved.

Thank you for considering my input to your research.
Rémy Asselot

**References**

Anderson, W., Gnanadesikan, A., Hallberg, R., Dunne, J., and Samuels, B. (2007). Impact of ocean color on the maintenance of the Pacic Cold Tongue. Geophysical Research Letters, 34(11).

Asselot, R., Lunkeit, F., Holden, P. B., & Hense, I. (2022). Climate pathways behind phytoplankton-induced atmospheric warming. *Biogeosciences*, *19*(1), 223-239.

Lengaigne, M., Madec, G., Bopp, L., Menkes, C., Aumont, O., and Cadule, P. (2009). Bio-physical feedbacks in the Arctic Ocean using an Earth system model. Geophysical Research Letters, 36(21).

Löptien, U., Eden, C., Timmermann, A., and Dietze, H. (2009). Effects of biologically induced differential heating in an eddy-permitting coupled ocean-ecosystem model. Journal of Geophysical Research: Oceans, 114(C6).

Manizza, M., Le Quéré, C., Watson, A. J., and Buitenhuis, E. T. (2005). Bio-optical feedbacks among phytoplankton, upper ocean physics and sea-ice in a global model. Geophysical Research Letters, 32(5).

Nakamoto, S., Kumar, S. P., Oberhuber, J. M., Ishizaka, J., Muneyama, K. and Frouin, R. (2001). Response of the equatorial Pacific to chlorophyll pigment in a mixed layer isopycnal ocean general circulation model. *Geophysical Research Letters*, 28(10), 2021-2024.

Oschlies, A. (2004). Feedbacks of biotically induced radiative heating on upper-ocean heat budget, circulation, and biological production in a coupled ecosystem-circulation model. Journal of Geophysical Research: Oceans, 109(C12).

Paulsen, H., Ilyina, T., Jungclaus, J. H., Six, K. D., and Stemmler, I. (2018). Light absorption by marine cyanobacteria affects tropical climate mean state and variability. Earth System Dynamics, 9(4):1283-1300.

Tian, F., Zhang, R. H., and Wang, X. (2021). Coupling ocean–atmosphere intensity determines ocean chlorophyll-induced SST change in the tropical Pacific. Climate Dynamics, 1-21.

---

## Author Comment (AC1)

Review of "How does the phytoplankton-light feedback affect the marine $N_2O$ inventory?" by Berthet et al.

**Rewiever #1**: Rémy Asselot, 25 Aug 2022

Summary

The phytoplankton-light feedback (PLF) is the mechanism defining the absorption of shortwave radiation by phytoplankton in the ocean. This mechanism affects the distribution of light in the water column and thus the oceanic temperature. As a consequence, the oceanic heat content and the sea-air gas fluxes are affected, altering the whole climate system.
This paper presents simulations with a complete or incomplete representation of PLF and describes ocean physics and biogeochemistry accordingly. To answer this question, the authors use an Earth system model with oceanic, sea-ice and marine biogeochemical components only. What is novel and interesting is that (1) depending on the representation of PLF, the climate system can be highly modified, and (2) simulated N2O budget are highly uncertain. I believe that this study deserves to be published and provides implications for future climate models development. However, before accepting this manuscript, I think that there are some aspects that could be improved. Especially the description of the model setup and the description of the processes explaining the results. I may have misread some things but I hope that the following comments will help the authors to improve their manuscript.

Our sincerest appreciation to you for your valuable comments and suggestions on our manuscript. We have addressed all of the points your raised in our revised manuscript, and we hope that the manuscript is now ready for publication. Please find below our detailed response (blue text) to your comments (black text). In our responses, changes to the manuscript are indicated by ***bold italicized*** text and lines numbers are relative to the new manuscript.

Please note that, as proposed by the second reviewer, simulation names have been changed in the new manuscript:
- chl_inter is now called REF
- clim_zcst is now called climZCST
- clim_zvar is now called climZVAR

Major comments:

Section 1

The authors seem to say that a consensus exists on the effect of PLF on the thermal structure of the ocean (line 96-102). However, several studies report an increase in SST due to PLF (e.g. Oschlies, 2004; Anderson et al., 2007; Lengaigne et al., 2009) while others report a cooling of the surface of the ocean (e.g. Nakamoto et al., 2001; Manizza et al., 2005; Löptien et al., 2009; Paulsen et al., 2018). So far, no consensus exists but the cooling effect reported in these studies might be due to the non- or weak coupling between the oceanic and atmospheric components of the models (Tian et al., 2021).

Thank you for this remark. In lines 99-104 we do not suggest that there is a consensus on the sign of the perturbation (SST cooling or SST warming), but rather that the first order of the PLF is to perturb the ocean thermal structure. But indeed in the previous submitted version there was no clear mention of cooling cases, which could lead to confusion. This has now been added and we hope that the new version reflects more clearly the state of the art on this issue. We modify the paragraph in the revised manuscript as follows (lines 99-117):

"Despite the diversity of modelled responses, a consensus emerges on the first order effect of PLF on the ocean physics, which is to perturb the ocean thermal structure (Nakamoto et al., 2001; Murtuggude et al., 2002; Oschlies, 2004; Manizza et al., 2005, 2008; Anderson et al., 2007; Lengaigne et al., 2007; Gnanadesikan and Anderson, 2009; Löptien et al., 2009; Patara et al., 2012; Mignot et al., 2013; Hernandez et al., 2017). By trapping more heat at the ocean surface in eutrophic regions, such as coastal or equatorial upwellings areas, the presence of phytoplankton *initially* increases the surface warming. Confining heat at the surface leads to less heat penetrating in subsurface. *In some cases, the advection and upwelling of subsurface cold anomalies can lead to remote cooling effects (Hernandez et al., 2017; Echevin et al., 2022). Dynamic readjustment in response to perturbations in thermal structure has also been shown to have a cooling effect, by increasing upwelling of cold water to the ocean surface. (Manizza et al. 2005; Marzeion et al., 2005; Nakamoto et al., 2001; Löptien et al., 2009; Lengaigne et al., 2007; Park et al., 2014).* Because these effects depend on upper ocean stratification, an important role is attributed to modelled seasonal deepening of the mixed layer as it determines the intensity of the underlying temperature anomaly and its vertical movement to the surface. In other terms, whatever the temporality of the causal chain, changes in the PLF representation are expected to both perturb the ocean heat uptake, and trigger perturbations of both the water column stratification and associated ocean dynamics."

Section 2.a

I understand that model configuration has already been described in Berthet et al. (2019) but it would be appropriate to give more explanation on the model configuration for the readers not familiar with Berthet et al. (2019). For instance, what is the vertical resolution of the oceanic grid? What is the depth of the first oceanic layer?

We have added some additional elements, in particular as suggested on the question of vertical discretization which is indeed important for solar flux penetration issues. But we would prefer not to go into details of the numerical choices at the risk of being too repetitive with Berthet et al. Lines 176-180 we added: "*Our modelled ocean has 75 vertical levels and the first level is at 0.5 meter depth. Vertical levels are unevenly spaced with 35 levels being in the first 300 meters of depth. Atmospheric forcings of momentum, incoming radiation, temperature, humidity, and freshwater are provided to the ocean surface by bulk formulae following Large and Yeager (2009).*"

Is phytoplankton simulated only in the first oceanic layer or can they go further down the water column?

Phytoplankton concentration is simulated along the 75 levels of our numerical ocean. PISCESv2 (Pelagic Interactions Scheme for Carbon and Ecosystem Studies volume 2) is a 3D

biogeochemical model which simulates the lower trophic levels of marine ecosystems (nanophytoplankton, diatoms, microzooplankton and mesozooplankton), the biogeochemical cycles of carbon and of the main nutrients (P, N, Fe, and Si). The revised manuscript clarifies this as follows (lines 199-201):

"*PISCESv2 simulates prognostic 3D distributions of nanophytoplankton and diatom concentrations. The evolution of phytoplankton biomasses is the net outcome of growth, mortality, aggregation and grazing by zooplankton (Aumont et al., 2015).*"

A comprehensive presentation of the model is found in Aumont et al. (2015).

Additionally, I couldn't really understand the description of the simulations from the text. However, from Fig.1 it is clear that the authors run a spin-up for 2000 years and then run their simulations for  20 years only. In total, they run their model for 2018 years. Is the model in steady state?

2000 years correspond to the accumulated spin-up of all the simulations we started from the older ones in our modelling group. We mentioned it in the first draft to certify that our model was in steady state. We agree that the presentation of the simulation protocol can be improved. To clarify we rewrote this part as follows (lines 184-190):

"*JRA55-do atmospheric reanalysis (Tsujino et al., 2018; Tsujino et al., 2020) provided the atmospheric forcings of the ocean. The global domain was first spun-up under preindustrial conditions during several hundred years ensuring that all fields approached a quasi-steady state. The historical evolution of atmospheric $CO_2$ and $N_2O$ concentrations was prescribed since 1850. To avoid the warming jump between the end of the spin-up and the onset of the reanalyses in 1958, the first 5 years of JRA55-do forcings were cycled, followed by the complete period of JRA55-do atmospheric forcing from 1958 to 2018.*"

Maybe the authors should also state that their results are the average of the last 10 years of the simulations.

We agree and we added the following sentence in section 2c (lines 281-282):
"*In the following temporal means cover the last 10 years of simulations, from 2009 to 2018. In other analyses the whole simulated period is shown (1999-2018).*"

In their simulations, do they prescribe the atmospheric CO2 and N2O concentrations or do they prescribe atmospheric CO2 and N2O emissions?

This point has been clarified in the revised manuscript at lines 187-188: "*The historical evolution of atmospheric $CO_2$ and $N_2O$ concentrations was prescribed since 1850*".

Does the ecosystem modelled consider only bulk phytoplankton or does it consider also e.g., cyanobacteria, diatoms? Is phytoplankton growth limited by light, nutrient and temperature?

PISCESv2 considers nanophytoplankton and diatoms. Phytoplankton biomasses experience growth, mortality, aggregation and grazing by micro- and mesozooplankton. Growth rate mainly depends on the length of the day, the depth of the mixed layer and the depth of the euphotic zone (defined as the depth at which there is 1‰ of surface photosynthetic available radiation). Light absorption by phytoplankton depends on the waveband and on the species. Normalized coefficients have been computed for each phytoplankton group by averaging and normalizing, for each waveband, the absorption coefficients published in Morel and Maritorena (2001). Nanophytoplankton growth depends on the external nutrient concentrations in N and P (Monod-like parameterizations of N and P limitations), and on Fe limitation which is modeled according to a classical quota approach. The production terms for diatoms are defined as for nanophytoplankton, except that the limitation term also include dissolved silica. Nutrient half-saturation constants vary with phytoplankton biomass of each compartment based on observations showing the increase in biomass to be due to the addition of larger size classes of phytoplankton. The aggregation term depends on the shear rate, as the main driver of aggregation is the local turbulence. The diatom aggregation term is increased in case of nutrient limitation because diatom cells are reported to excrete mucus under nutrient stress which increases their stickiness.

We expanded the model description in the revised manuscript (lines 199-202):

"*PISCESv2 simulates prognostic 3D distributions of nanophytoplankton and diatom concentrations. The evolution of phytoplankton biomasses is the net outcome of growth, mortality, aggregation and grazing by zooplankton (Aumont et al., 2015). Light absorption by phytoplankton depends on the waveband and on the species (Bricaud et al., 1995).*"

I think it would also be good to give the absorption coefficients used to parameterize PLF or directly give the equation of PLF.

We agree, thanks. This is now indicated in the manuscript lines 202-215:

"*A simplified formulation of light absorption by the ocean is used in our experiments to calculate both the phytoplankton light limitation in PISCESv2 and the oceanic heating rate (Lengaigne et al, 2007). In this formulation, visible light is split into three wavebands: blue (400–500 nm), green (500–600 nm) and red (600–700 nm); for each waveband, the CHL-dependent attenuation coefficients, $k_R, k_G$ and $k_B$, are derived from the formulation proposed in Morel and Maritorena (2001):*"

$$k_{WLB} = \sum_{\lambda_1}^{\lambda_2} \left( k(\lambda) + \chi(\lambda)[\text{CHL}]^{e(\lambda)} \right)$$

"*where WLB means the wavelength band associated to red (R), green (G) or blue (B), and bounded by the wavelengths $\lambda_1$ and $\lambda_2$ as detailed above. $k(\lambda)$ is the attenuation coefficient for optically pure sea water. $\chi(\lambda)$ and $e(\lambda)$ are fitted coefficients which allows to determine the attenuation coefficients due to chlorophyll pigments in sea water (Morel and Maritorena, 2001).*"

**Section 2.b**

The paragraph line 240-252 would better fit in the introduction.

To comply with the demand of the other reviewer we moved the whole section 2b to the supplementary material. We choose to keep the paragraph describing N2O production in our "N2O parameterization" section (now in supplementary) because it highlights how the model parameterization represents N2O processes compared to what happens in the real ocean. The "N2O part" of our introduction aims to explain the spatial coincidence between N2O emissions regions and CHL, as well as the current uncertainties in N2O modelling, in order to introduce the need to explore how the PLF may affect marine N2O inventory. We do think that moving the paragraph describing N2O production to the introduction could sidetrack the reader from our main message.

**Section 3.a**

I would suggest to put the chlorophyll maps in the results section rather than in appendix because it helps to understand to results. Especially, PLF affects the OHC via chlorophyll concentration so it might be easier to follow and easier to understand if the chlorophyll concentration maps are directly in the results section.

Agreed, Figure S1 has been transferred in the main text as Figure 2.

Furthermore, it seems that the authors use their two simulations with incomplete representation of PLF (chl_zcst and chl_zvar) as upper and lower limit of their uncertainties (line 403-405). I wonder why they think that?

In fact, we did not use them arbitrarily as upper/lower bounds. Fig 1 shows that time series of OHC300 from climZCST and climZVAR surround our control experiment: "However Figure 1 highlights that the simulation using a consistent CHL for interacting with both incoming SW and biogeochemical cyclings (REF) does not amplify one of these two trends, as climZCST and climZVAR surround REF". The analysis revealed them as upper/lower bounds for ocean heat content and we used them as such.

Do they consider that chl_zcst is a simple simulation giving the minimum OHC300 while they consider chl_zvar as a complex simulation (close to reality) giving the "real" OHC300? And that REF is somehow fluctuating between these two simulations?

No, due to their use of an incomplete PLF, we consider that both climZCST and climZVAR simulations are more simple configurations than our control run REF. REF is the complex simulation.

**Section 3.b**

As in Sweeney et al. (2005), the authors state that small changes in chlorophyll concentration drives important changes of the mixed layer depth in the subtropical regions (line 447-453).

However, they do not explicitly explain what are the mechanisms behind these changes in MLD. Is it due to PLF? What is the link between chlorophyll and MLD? I think what the authors want to say is that changes in chlorophyll drive changes in temperature via PLF, which in turn drives changes in thermocline/pycnocline and thus in MLD.

Yes in section 3a we first show that changes in CHL drive changes in temperature and ocean heat content. Next, Figure 4 illustrates that these thermal perturbations are associated to dynamical ones. Finally in section 3b we insist on the fact that the heat and subsequent dynamical modifications are associated to pycnocline perturbations (lines 384-385): "Perturbations of the annual pycnocline depth (Figure 5, a-c) highlight a vertical adjustment to the heat (Figure S2) and subsequent large-scale dynamical anomalies (Figure 4)." It is our assumption that perturbations of the isopycnal layers detected in the pycnocline/thermocline (Figure 5) give evidences of MLD perturbations. This result is consistent with the MLD perturbations due to small CHL changes in subtropical gyres highlighted by Sweeney et al. (2005).

The authors state that the decrease/increase in N2O concentration is driven by different mechanisms depending on the region studied. For instance, the decrease in N2O concentration in the South Pacific is due to increased temperature, enhanced circulation and deepening of the pycnocline. But in the North Atlantic the increase in N2O concentration is only due to the shoaling of the pycnocline. Additionally, in the North Pacific, the decrease in N2O concentration is due to the higher O2 concentration. Why do we have different mechanisms involved at different places of the world? For instance, why does O2 concentration is important for the North Pacific region but not for the North Atlantic or South Pacific regions?

Equation 3 in section A of the supplementary material allows to understand the different mechanisms involving N2O production (remineralization, grazing and nitrification) and depletion. These mechanisms vary in space and time, and do not occur at all places of the 3D ocean with the same intensity. In addition, the local suboxic production of [N2O] is adapted to the local oxygen concentration (through the f(o2) function, please refer to equation 1). This explains the specific regime seen in the oxygen minimum zone of the North Pacific.

N2O concentration reflects both local and advected production/depletion. Depending on the region the role of the circulation differs. The goal of this paragraph is to show that N2O concentration is driven by regional features which are a combination of transport, stratification and biogeochemical processes.

Section 4

In conclusion, the authors detail some results about oxygen without showing them previously. I think they should talk about these results in the result section rather than showing them suddenly in the conclusion section.

We consider that the paragraph about OMZ volume is a discussion which extends the significance of our study. In that perspective we renamed the last section "Discussion and conclusion". (Note that we replaced all OMZ shorthand notations by the whole expression, as asked by the second reviewer).

Furthermore, the authors should also discuss the fact that their model setup does not consider an atmospheric component. Do they expect similar results if they use a coupled ocean-atmosphere model rather than using an atmospheric N2O and CO2 forcing? For instance, Asselot et al. (2022) show that PLF affects the climate system mainly via sea-air greenhouse gas fluxes. Adding an interactive atmospheric component could therefore lead to higher greenhouse gases in the atmosphere, increasing the atmospheric temperature. The higher atmospheric temperature might increase the oceanic temperature and thus enhance the effect of PLF on N2O fluxes.

We agree that the use of a forced ocean model may limit the response. This issue is now discussed, with the following paragraph that has been added to our conclusion section (lines 515-523):
"*In forced ocean simulations, atmospheric forcings constrain surface temperature, salinity and thus solubility. However, the N2O concentration integrated over the upper 300 meters depth of the water column (Figure 5, e-f) showed differences with the control run that follow those of the in-depth temperature (Figure S2, c-d): in climZCST (climZVAR), a warmer (colder) tropical ocean leads to a decreased (an increased) N2O concentration. Because higher marine greenhouse gas emissions will increase the temperature of the coupled atmosphere-ocean system, adding an interactive atmospheric component is expected to amplify the PLF-induced mean changes in marine N2O concentration highlighted in this ocean-only numerical set (Park et al., 2014; Asselot et al., 2022)*."

Specific comments

Line 17: Replace "thanks to" by "by".
Done.

Line 22: Replace "experiments" by "simulations".
This comment is valid for the entire manuscript.
Done.

Line 23: Replace "have been performed" by "are performed".
The sentence has been rephrased as follows (line 23): "*We exploit global sensitivity simulations at 1-degree of horizontal resolution over the last two decades (1999-2018) coupling ocean, sea ice and marine biogeochemistry*".

Line 34-35: Replace "shine a light on a current uncertainty of the modelled marine nitrous oxide budget in that climate models." by "shine light on current uncertainties of modelled marine nitrous oxide budget in that climate model."
We rephrased as follows (line 34): "*Our results based on a global ocean-biogeochemical model at CMIP6 state-of-the-art shed light on current uncertainties in modelled marine nitrous oxide budgets in climate models*".

Line 41: Replace "suffers" by "undergoes"
We rephrased as follows (line 41): "*This natural effect is either not represented in the ocean component of climate models, or included in a simplified manner*".

Line 43: Replace "uncertains the forecast" by "leads to uncertain forecast"
We rephrased as follows (line 44): "...*which in turn leads to uncertainties in projections of oceanic emissions...*".

Line 92-94: The authors state that "Two main causal chains have been proposed to interpret the sign of the final heat perturbation" but they do not give these two causal chains.
That was the idea of the rest of the sentence. We rewrote the paragraph as follows (lines 94-98): "Two main causes were put forward to explain the sign of the final heat perturbation: *either an indirect dynamical response (Murtugudde et al., 2002; Löptien et al., 2009) or a direct thermal effect (Mignot et al., 2013; Hernandez et al., 2017). Hernandez et al. (2017) further distinguished a local from a remote thermal effect by highlighting the important role played by the advection of offshore CHL-induced cold anomalies in the Benguela upwelling waters.*"

Furthermore "causal chains" could be directly replaced by "causes".
Done.

Line 132: I guess the authors are speaking about atmospheric emissions here.
No we are still discussing oceanic emissions. As written just before, 20% of the annual N2O flux occurs in the coastal upwelling systems that are undersampled by observations. Figure 1A of Yang et al. (2020) presents an up-to-date map of spatial distributions.

Line 132: Which decade?
In the revised manuscript we specified " The recent global budget of Tian et al. (2020) estimates natural sources from soils and oceans to contribute with up to 57% to the total $N_2O$ emissions *between 2007 and 2016*, with the ocean flux reaching 3.4 (2.5–4.3) Tg N yr$^{-1}$."

Line 136: Replace "model marine" by "simulate marine"
Done.

Line 157: Parameterization.
The sentence has been deleted because the section has been moved to the supplementary material.

Line 202-203: Rephrase please
Done. "These two simulations differ from each other by the "realism" of the vertical profile derived *in each grid point* from the *surface value of the* ESACCI CHL climatology *to the level of light extinction* (Table 1). *climZCST uses constant profiles of CHL spreading uniformly in the vertical direction (Figure 2, b and d-f). climZVAR uses variable vertical profiles computed following Morel and Berthon (1989) (Figure 2, c and d-f)*."

Line 213: Replace "biogeochemical element cycling" by "biogeochemical cycles".
Done .

Line 325: Replace "by the Ifremer" by "by Ifremer".
Done.

Line 332-337: The authors compare their modelled temperature and oxygen with several database. Is it shown somewhere? In figures or appendix?
Yes, in figure 3 (temperature), and supplementary figures S2 (temperature) and S5 (oxygen).

Line 388-392: I cannot say what the authors mean with these two sentences.

The sentence has been rephrased as follows (line 334): *"It can be expected that experiments having spin-ups run with different representations of the PLF, would give even stronger sensitivities than those highlighted in this study. The sensitivities of OHC300 to the PLF formulation evaluated here should be considered at the lower end of estimate of OHC discrepancies that may emerge from changing the PLF representation."*

In other words, we may expect that a simulation in which even the spin-up period would have been run with an incomplete PLF will show much greater difference in OHC300 with our control simulation (REF), as the deviation increased along time (and spin-up period encompasses a long period).

Line 393: green on Figure 1?
Yes that was the initial idea, but colors indications have been deleted from the text to improve readability.

Line 394: directly use OHC300 as it is previously used.
Done.

Line 400: blue on Figure 1?
Yes that was the initial idea, but color indications have been deleted from the text to improve readability.

Line 427: Replace "modifying" by "different".
Done.

Line 442: Replace "raising" by "shallowing".
It is true that it corresponds to a "shallowing of the isopycnals", but what we want to highlight here is their rise to the surface. We reformulated as follows (line 389): "as it triggers an upward displacement of the isopycnals ".

Line 445-447: "Over these … the pycnocline" please rephrase.
We rephrased by splitting the sentence as follows (line 392): "Over these subtropical gyres heat is redistributed along the vertical as the subsurface warm anomaly dives. *The subduction of these heat anomalies causes* in turn a deepening of the pycnocline (**Error! Reference source not found.**, b and c)."

Line 460: Replace "upper line" by "upper panel".
Done.

Line 466-469: Please rephrase.
The sentence has been deleted.

Line 474: Replace "By contrast" by "In contrast".
Done.

Line 484: Remove "next".
Done.

Line 484-486: "Approaching … heat uptake" please rephrase.
For clarity we splitted the sentence as follows (line 428):
"The relationship between $N_2O$ concentration and OHC300 in the Tropical Ocean is derived from a linear regression for each of the three 20-years simulations (Error! Reference source not found.Figure 6). *The resulting slopes allow to identify three distinct tropical $N_2O$ production pathways along time as a function of the oceanic heat uptake.*"

Line 500: Should be section 3.c rather than section 3.d
Done.

Line 502: Replace "the degree of realism of the PLF" by "the way PLF is simulated/modelled".
Done.

Line 504-507: Please rephrase because it seems that you consider that Dpn2o anomalies reflect only differences in surface N2O concentration while Dpn2o anomalies can also reflect differences in N2O solubility.
Solubility is mainly driven by temperature and salinity. Figures 3 and S3 show that the majority of temperature and salinity anomalies in climZCST and climZVAR compared to REF are located below 25 meters depth. We have almost no solubility perturbations close to the surface. This explains why spatial patterns of Dpn2o perturbations are similar to that of surface N2O concentration. To clarify we inserted a new sentence at lines 449:

"Because the atmospheric partial pressure of N2O is identical among simulations, differences in Dpn2o are driven by changes in surface N2O concentration normalized by those in N2O solubility. *Since solubility is mainly driven by temperature and because surface temperature anomalies are very weak (Figure S3, c and d), we do not expect solubility perturbations close to the surface.* It results that spatial patterns of Dpn2o anomalies (Error! Reference source not found.) reflect differences in surface N2O concentration."

Line 512-513: Are the units atm or natm?
natm, thank you for detecting the typo.

Line 548: Remove "and".
Done.

Line 551: Replace "add" by "added".
Done.

Line 552: Remove ", and".

Done.

Line 552: Remove "with time".
We replace "with" by "along".

Line 556: Replace "their" by "a".
Done.

Line 591: Replace "so why it used" by "that is why it uses"
Done.

Figure S4: I don't understand how densities (y-axis) can be negative. Is it a density anomaly represented?
Yes, densities are plotted as anomalies compared to 1999. The following sentence has been added to the legends of Figures 6 and S4 in the revised manuscript: "*All points reflect anomalies compared to year 1999*".

Figure 3a: The colorbar can be improved.
Done.

Thank you for considering my input to your research.
Rémy Asselot

**References**

Anderson, W., Gnanadesikan, A., Hallberg, R., Dunne, J., and Samuels, B. (2007). Impact of ocean color on the maintenance of the Pacic Cold Tongue. Geophysical Research Letters, 34(11).

Asselot, R., Lunkeit, F., Holden, P. B., & Hense, I. (2022). Climate pathways behind phytoplankton-induced atmospheric warming. Biogeosciences, 19(1), 223-239.

Lengaigne, M., Madec, G., Bopp, L., Menkes, C., Aumont, O., and Cadule, P. (2009). Biophysical feedbacks in the Arctic Ocean using an Earth system model. Geophysical Research Letters, 36(21).

Löptien, U., Eden, C., Timmermann, A., and Dietze, H. (2009). Effects of biologically induced differential heating in an eddy-permitting coupled ocean-ecosystem model. Journal of Geophysical Research: Oceans, 114(C6).

Manizza, M., Le Quéré, C., Watson, A. J., and Buitenhuis, E. T. (2005). Bio-optical feedbacks among phytoplankton, upper ocean physics and sea-ice in a global model. Geophysical Research Letters, 32(5).

Nakamoto, S., Kumar, S. P., Oberhuber, J. M., Ishizaka, J., Muneyama, K. and Frouin, R. (2001). Response of the equatorial Pacific to chlorophyll pigment in a mixed layer isopycnal ocean general circulation model. Geophysical Research Letters, 28(10), 2021-2024.

Oschlies, A. (2004). Feedbacks of biotically induced radiative heating on upper-ocean heat budget, circulation, and biological production in a coupled ecosystem-circulation model. Journal of Geophysical Research: Oceans, 109(C12).

Paulsen, H., Ilyina, T., Jungclaus, J. H., Six, K. D., and Stemmler, I. (2018). Light absorption by marine cyanobacteria affects tropical climate mean state and variability. Earth System Dynamics, 9(4):1283-1300.

Tian, F., Zhang, R. H., and Wang, X. (2021). Coupling ocean–atmosphere intensity determines ocean chlorophyll-induced SST change in the tropical Pacific. Climate Dynamics, 1-21.

The comment was uploaded in the form of a supplement:
https://esd.copernicus.org/preprints/esd-2022-28/esd-2022-28-RC1-supplement.pdf

Citation: https://doi.org/10.5194/esd-2022-28-RC1

---

## Author Comment (AC2)

Review of "How does the phytoplankton-light feedback affect the marine $N_2O$ inventory?" by Berthet et al.

**Reviewer #2**: Anonymous Referee #2, 16 Sep 2022

OVERVIEW

The authors study the effect of light absorption by phytoplankton on the heat content of the ocean and subsequently on N20. They run simulations with full/partial representation of this process and study its effect. They demonstrate that indeed the biooptical coupling influences N20 and the heat content of the ocean. I propose some modification to this manuscript prior to publication, which are given below.

We appreciate your thoughtful review of our manuscript. We have addressed all points you raised in our revised manuscript, and we hope that the manuscript is now ready for publication. Please find below our detailed response (blue text) to your comments (black text). In our responses, changes to the manuscript are indicated by ***bold italicized*** text and lines numbers are relative to the new manuscript.

GENERAL COMMENTS

1.  The model was spun up considerably longer than the simulation runs. Please elaborate as to why not use longer simulation runs?

Our framework shows that a period of 20 years is long enough to highlight that the 3 numerical solutions diverge (see Figure 1). We choose the period 1999-2018, as it corresponds to the last 20 years of the JRA55 atmospheric forcing which is the most realistic forcing we currently have. In addition, this period is of interest because it is well covered by observations: for example, observations of ocean heat content really started in 2005 with a massive Argo deployment.

In the first version of the manuscript we mentioned that the model was spun-up for 2000 years. This correspond to the accumulated spin-up of all the simulations that we started from the older ones in our modelling group. We mentioned it in the first draft to certify that our model was in quasi-steady state. But we agree that the protocol was not very understandable for the reader. To clarify our spin-up protocol we rewrote this part as follows (lines 184-190):

"*JRA55-do atmospheric reanalysis (Tsujino et al., 2018; Tsujino et al., 2020) provided the atmospheric forcings of the ocean. The global domain was first spun-up under preindustrial conditions during several hundred years ensuring that all fields approached a quasi-steady state. The historical evolution of atmospheric $CO_2$ and $N_2O$ concentrations was prescribed since 1850. To avoid the warming jump between the end of the spin-up and the onset of the reanalyses in 1958, the first 5 years of JRA55-do forcings were cycled, followed by the complete period of JRA55-do atmospheric forcing from 1958 to 2018.*"

2.  The overall text seems to be missing a thread and has too many shorthand notations. While familiar to the authors and others in the field, this hampers the flow of the paper

substantially. I suggest reducing the usage of shorthand notation and move some of the technical parts to the supplement.

We reduced the use of shorthand notation by suppressing six of them (SW, SST, OMZ, ORCA1, T/S and WOA09). We only kept the ones that we use all along the text: PLF (phytoplankton-light feedback), OHC300 (ocean heat content of the upper 300 meters depth), CHL (chlorophyll), Dpn2o (N2O partial pressure difference across the air-sea interface) and chemical names (N2O/CO2/O2). NEMO, PISCESv2 and CNRM-ESM2-1 are model names. CMIP and OMIP are acronyms very used in climate sciences, but to clarify we added the definition of CMIP6 in the abstract at lines 16: " Only one third of the Earth system models contributing to *the 6th phase of the Coupled Model Intercomparison Project (CMIP6)* includes a complete representation of the PLF."

As requested section 2b presenting the N2O parameterization has been moved to the supplementary material.

3. Given the paper deals with the effect of the biooptical coupling I suggest the authors add exact mathematical expressions which describe this process. It is described somewhat shortly in lines 218 to 224. For example: please state the exact equations which model how phytoplankton affects light penetration.

Please see how we reorganize section 2 ("Methodology"). While the subsection presenting the N2O parameterization has been moved to the supplementary material, a new subsection (2b) is now dedicated to the formulation of the PLF, in which we describe it as follows (starting at line 192):

"b) Experimental design: three representations of the PLF

The control simulation (hereafter REF) together with the spin-up both account for a fully interactive PLF: the penetration of shortwave radiation into the ocean surface is constrained by the CHL concentration ([CHL]) produced by the PISCESv2 biogeochemical component *(Figure S1, REF)*.

*PISCESv2 simulates prognostic 3D distributions of nanophytoplankton and diatom concentrations. The evolution of phytolankton biomasses is the net outcome of growth, mortality, aggregation and grazing by zooplankton (Aumont et al., 2015). Light absorption by phytoplankton depends on the waveband and on the species (Bricaud et al., 1995). A simplified formulation of light absorption by the ocean is used in our experiments to calculate both the phytoplankton light limitation in PISCESv2 and the oceanic heating rate (Lengaigne et al., 2007). In this formulation, visible light is split into three wavebands: blue (400–500 nm), green (500–600 nm) and red (600–700 nm); for each waveband, the CHL-dependent attenuation coefficients, $k_R, k_G$ and $k_B$, are derived from the formulation proposed in Morel and Maritorena (2001):*

$$k_{WLB} = \sum_{\lambda_1}^{\lambda_2}\big(k(\lambda) + \chi(\lambda)[CHL]^{e(\lambda)}\big) \qquad (1)$$

*where WLB means the wavelength band associated to red (R), green (G) or blue (B), and bounded by the wavelengths $\lambda_1$ and $\lambda_2$ as detailed above. $k(\lambda)$ is the attenuation coefficient*

*for optically pure sea water. $\chi(\lambda)$ and $e(\lambda)$ are fitted coefficients which allows to determine the attenuation coefficients due to chlorophyll pigments in sea water (Morel and Maritorena, 2001).*

At year 1999 two sensitivity experiments were branched off (**Error! Reference source not found.**). Both simulations climZCST and climZVAR account for an incomplete and external PLF, as they consider an observed climatology of surface [CHL] from ESACCI (Valente et al., 2016) in order to compute the light penetration into sea water (*Equation 1; Figure S1*). These two simulations differ from each other by the "realism" of the vertical profile derived in each grid point from the surface value of the ESACCI CHL climatology to the level of light extinction (Table 1). *climZCST uses constant profiles of CHL spreading uniformly in the vertical direction (Figure 2, b and d-f). climZVAR uses variable vertical profiles computed following Morel and Berthon (1989) (Figure 2, c and d-f)*. This set of simulations is representative of the several configurations used in the case of CMIP intercomparison project.

[...]"

Subsequently, how does this affect the rate of phytoplankton growth, rate of heating at depth, and so on. I consider this to be of high value for non-experts. Also, it would make the model more easily reproducible.

As shown by equation 1 the PLF has only one direct effect, on the quantity of light penetrating the water column. By modulating the quantity of light reaching the subsurface, the chlorophyll-light interaction will effectively change the ocean heat content and the photosynthesis. However, as explained in the introduction, the net effect of PLF is a mix between direct thermal (local and remote) and indirect dynamical effects. This is why there is no simple quantification of the perturbation that the PLF exerts on phytoplankton growth or rate of heating at depth. To quantify these effects our only solution is to integrate a 3D model of ocean physics and marine biogeochemistry.

4.   Please state the governing equations for phytoplankton dynamics: in particular the growth term of phytoplankton as a function of light, nutrients and temperature. What limits growth? How is the loss term parametrized? Does light penetration feedback onto the growth rate? Does temperature effect the growth rate? Is the model time step adequate to resolve this?

PISCESv2 (Pelagic Interactions Scheme for Carbon and Ecosystem Studies volume 2) is a 3D biogeochemical model which simulates the lower trophic levels of marine ecosystems (nanophytoplankton, diatoms, microzooplankton and mesozooplankton), the biogeochemical cycles of carbon and of the main nutrients (P, N, Fe, and Si). The revised section 2b clarifies this as follows (lines 199-201):

"*PISCESv2 simulates prognostic 3D distributions of nanophytoplankton and diatom concentrations. The evolution of phytoplankton biomasses is the net outcome of growth, mortality, aggregation and grazing by zooplankton (Aumont et al., 2015).*"

Aumont et al. (2015) give a complete description of this complex biogeochemical model which represents the evolution of 24 prognostic variables (see Figure R1) and in which 5 pages are dedicated to phytoplankton dynamics. We cannot reproduce them here, but we give you a short overview below:

PISCESv2 considers that biomasses of nanophytoplankton and diatoms experience growth, mortality, aggregation and grazing by micro- and mesozooplankton (see attached Figure R1). The 3 latter processes compose what you called "the loss term". Growth rate mainly depends on the length of the day, the depth of the mixed layer and the depth of the euphotic zone (defined as the depth at which there is 1‰ of surface photosynthetic available radiation). Light absorption by phytoplankton depends on the waveband and on the species. Nanophytoplankton growth depends on the external nutrient concentrations in N and P (Monod-like parameterizations of N and P limitations), and on Fe limitation which is modeled according to a classical quota approach. The production terms for diatoms are defined as for nanophytoplankton, except that the limitation terms also include Silicate. Nutrient half-saturation constants vary with the phytoplankton biomass of each compartment because the observations show that the increase in biomass is generally due to the addition of larger size classes of phytoplankton. The aggregation term depends on the shear rate, as the main driver of aggregation is the local turbulence. The diatoms aggregation term is increased in case of nutrient limitation because it has been shown that diatoms cells tend to excrete a mucus which increases their stickiness. As a consequence, collisions between cells yield to a more efficient aggregation process.

Modulating light penetration through the PLF is expected to change light availability for biogeochemical cycles (labelled "Photosynthetic Available Radiation" on Figure R1) as well as physical properties of the ocean waters (what we show in our manuscript). That would indirectly change the conditions for CHL production. However, our set of experiments show no marked perturbations in terms of CHL production in climZCST and climZVAR compared to REF (Figure R2). As shown by our manuscript, main biogeochemical perturbations arise from those of temperature, ocean heat content, dynamics and stratification.

Finally, yes, the time step is adequate to resolve the processes mentioned above on a spatial grid at 1 degree of horizontal resolution: our oceanic configuration of the NEMO model had a timestep of 1800 seconds (30 min) and marine biogeochemistry (PISCESv2) has been called at every timestep of the ocean physics.

[Figure]

Figure R1: Schematic diagram of the standard version of PISCESv2 describing the nutrients, carbon and oxygen cycles and processes modelled.

[Figure]

Figure R2: Time versus depth diagram of the mean chlorophyll (mg/m³; shading) and mixed layer thickness (m; color line) averaged over the tropical area [35S-35N; 0-360E].

Note that for our study the marine N2O cycle (now described in the supplementary material on your request) has been added in PISCESv2.

5. Please add a figure displaying the model structure, highlighting the part related to biooptical coupling and the link to nitrous oxide, as it is the core of the paper.

Figure R3 below has been added in the supplementary material, as Figure S1. We choose to not add it in the main text.

[Figure]

Figure R3: Schematic diagram of the numerical set. The phytoplankton-light feedback (PLF) encompasses the interaction between the incoming solar radiation (identical among the 3 simulations) and the CHL concentration used to filter its penetration into the water column. Different representations of the PLF are distinguished in function of the CHL used to filter the incoming radiation: it is either computed from PISCESv2 model (REF) or externally prescribed

from an observed climatology (climZCST and climZVAR). We show that the nature of the CHL chosen to interact with light determines different states of the ocean physics (e.g. OHC, temperature, dynamics, stratification) that drive different states of the marine biogeochemistry (e.g. N2O, CHL, O2).

6. The effect of phytoplankton on the mixed layer depth is mentioned in line 447 (I assume the authors mean heating due to the biooptical feedback). Here the effect of wind is stated to alter the mixed layer depth more strongly than phytoplankton. However, it is known since Platt et al. (2003) that the phytoplankton change their biomass so that the Critical depth (Sverdrup, 1953) matches the mixed layer depth. This occurs due to the biooptical coupling that the authors explore. Please elaborate more on this.

The winds and air-sea fluxes are expected to control the mixed-layer depth and more generally the thermal stratification more strongly than phytoplankton. The point we wanted to raise in the passage between line 395 and 399 was that the differential heating is expected to have a major impact in regions with low mixing conditions (in region with high mixing the effect of differential heating between the surface and subsurface layers is almost instantaneously mixed and is not expected to have some impact on the stratification). This interpretation is in line with Figure 5 that shows a major impact on the pycnocline depth in the low wind subtropical regions.

The corresponding sentence has been modified as follows: « *As stressed by Sweeney et al. (2005), small changes in CHL concentration (Figure 2) may have important effects on the mixed layer depth in these subtropical gyres due to low local wind speeds and low mixing conditions. This is thought to explain the large sensitivity we observe in terms of pycnocline depth (Figure 5) and ocean heat content in these regions.*"

Regarding the attraction of the critical depth to the mixed-layer depth. A first point we want to raise is that the CHL produced by the biogeochemical model PISCESv2 is poorly impacted by the change of PLF representation (Figure R2; and we show that N2O concentration is much more sensitive than CHL to these changes). This suggests that in average the critical depth is weakly impacted by the PLF. But we did not check whether the attraction of the critical depth to the mixed layer depth may be locally modulated by the winds/mixing conditions. This could be interesting but we think that it is out of the scope of our study.

REFERENCES

Sverdrup, H. U. (1953). On conditions for the vernal blooming of phytoplankton. *Journal du Conseil International Pour lExplorationde la Mer*, 18: 287–295.

Platt, T., Broomhead, D. S., Sathyendranath, S., Edwards, A. M., Murphy, E. J. (2003). Phytoplankton biomass and residual nitrate in the pelagic ecosystem. Proceeding of the Royal Society A, 459: 1063–1073.

SPECIFIC COMMENTS

L14 Please rephrase the sentence starting with "Considering…".

Done, see lines 14-15: "*The PLF allows to simulate differential heating across the ocean water column as a function of the phytoplankton concentration.*"

L65 Change "into" to "through".
The sentence has been reformulated (line 65): "*It implies that the chlorophyll (CHL) produced by the biogeochemical model is used to determine the fraction of shortwave radiation penetrating ocean surface waters.*"

L67 Please remove "(because the same)".
Done.

L72 – L109 Suggested references on this effect from the literature:
Edwards, A. M., Wright, D. G., Platt, T. (2004) Biological heating effect of a band of phytoplankton. *Journal of Marine Systems*, 49, 89-103. doi: 10.1016/j.jmarsys.2003.05.011.
Edwards, A. M., Platt, T., Wright, D. G. (2001) Biologically induced circulation at fronts. *Journal of geophysical research*, 49, 89-103. doi: 10.1016/j.jmarsys.2003.05.011.
Thanks for that interesting papers. We added the references to your papers at line 79 as follows:
"Enabling a phytoplankton-light interaction *modifies the hydrodynamics of the water column (Edwards et al., 2001; Edwards et al., 2004)*, the intensity of the spring-bloom in subpolar regions (Oschlies, 2004), the maintenance of the Pacific Cold Tongue (Anderson et al., 2007), the seasonality of the Arctic Ocean (Lengaigne et al., 2009), the strength of the tropical Pacific annual cycle, as well as the ENSO variability (Timmermann and Jin, 2002; Marzeion et al., 2005), the northward extension of the meridional overturning circulation (Patara et al., 2012) and the cooling of the Atlantic and Peru-Chili upwelling systems (Hernandez et al., 2017, Echevin et al., 2022)."

L152 Please rephrase the sentence starting with "In that perspective…".
We changed the sentence as follows (line 157): "*Here we investigate* how an incomplete representation of the *PLF leads to uncertainties in N₂O projection* in an up-to-date global ocean-biogeochemical model making up the current generation of Earth system models."

L209 Table 1 should be on top of the table. Suggest renaming chl_inter, clim_zcst, clim_zvar so as to drop the "_".

The legend of Table 1 has been moved to the top of the table.

We renamed our simulations and drop the underscore. Our control experiment which uses an interactive CHL to infer the PLF is now called "REF". Our two sensitivity experiments are now called climZCST and climZVAR. "clim" refers to the use of an externally prescribed climatology, and "ZCST" or "ZVAR" denote the nature of the vertical profile z that we respectively impose constant or variable.

L231 – L317 Please move to the Introduction.
Following your suggestion to move the technical parts to the supplementary, we moved section 2b regarding N2O parameterization to the supplementary material.

Citation: https://doi.org/10.5194/esd-2022-28-RC2

---

## Referee Report (RR1)

**Review 2 of Berthet et al. – How does the phytoplankton-light feedback affect marine N₂O inventory?**

The authors have carefully revised their manuscript in response to the reviewer comments, and I believe the manuscript has improved substantially as a result and is ready to be published with a few minor revisions. Below I focus on the responses to my own comments and on the revised manuscript. I just have on major comment concerning the steady-state of the model and if this comment is adequately addressed in the revised manuscript, the manuscript can be published.

**Major comments**

Looking at the response to my previous review and from the revised manuscript, the authors clearly state that their model is in steady-state. I understand that the spin-up is in steady-state after running for 2000 years but I really question the fact that after 20 years of simulations, the model simulations are in steady state. A good argument to say that their simulations are not in steady-state is Fig. 1. This figure shows an increase of OHC300 over the whole simulations (1999-2018). If the model would be in steady-state during the "simulations period", the OHC300 would be constant and not fluctuate anymore after a certain period of time. I think the simulations are too short (only 20 years) to reach a steady state. This point should be clearly stated in the manuscript.
Moreover, is PLF taken into account in the spin-up ?

**Minor comments**
The line numbers correspond to the tracked changes version of the manuscript.

Line 34-36: Please rephrase.

Line 41: Replace "effect" by "process".

Line 43: Replace "biophysical" by "biogeophysical" and same for the rest of the manuscript.

Line 44: Replace "which leads" by "leading".

Line 67-69: Replace "is consistent" by "is the same as".

Line 75: Replace "included to" by "included in".

Line 78: Replace "affected" by "affects".

Line 91: Replace "than to the" by "than the".

Line 92-93: Replace "amplifies the mean of PLF-induced changes, but without altering the sign" by "amplifies the magnitude of the PLF-induced changes, without altering the sign".

Line 107-108: is the remote cooling effect at the surface or at the subsurface ?

Line 122: 265-298 times higher ?

Line 257: Replace "consistent" by "identical".

Line 261-262: Please rephrase.

Line 285: Replace "the year" by "when".

Line 287: Replace "OHC of deeper layers" by "OHC in the deeper layers".

Line 297-298: I am not sure by these two lines. Are you results detailed following two different time periods ? One time period covers 10 years (2009-2018) and a second one covers a longer period (1999-2018) ?

Line 326: Replace "comparable to that of our" by "comparable to our".

Line 329: Replace "accordance" by "agreement".

Line 331: Replace "order" by "about".

Legend Figure 3: I would say that the 60°S-60°N is not the tropical band anymore. I would encompass that you calculate the average over the tropical and mid-latitudes band.

Line 335: Replace "branching of" by "switching on".

Line 338: Replace "weaker profiles" by "smaller/weaker concentrations".

Line 341: Replace "by comparison" by "compared".

Line 351: Replace "spin-ups run" by "spin-up runs".

Legend Figure S4: I am confused by the units. The potential density (y-axis) should be in kg/m$^3$ as in the main text (line 360). The OHC300 (x-axis) should be in ZJ and not ZJ/yr.

Line 360-362: The sentence can be reduced and directly state "The opposite trend (a reduced OHC300 compared to REF) is simulated when considering a variable vertical profile of CHL (climZVAR)."

Line 362-365: This sentence can be reduced as well, giving "However Figure 1 highlights that the simulation REF does not amplify one of these two trends, as climZCST and climZVAR surround REF."

Line 370: Replace "Ranges of uncertainty" by "uncertainty ranges".

Line 373-375: Please repharse.

Legend Figure 6: The units of $N_2O$ should be µmolN/m$^2$ and the units of OHC300 should be ZJ.

Line 446: Replace "along time" by "through time".

Line 449: Replace "appear a" by "appears as a".

Line 464: Please define Dpn2o as it is the first time it appears in the main text.

Line 466: I guess you mean surface oceanic $N_2O$ concentrations.

Line 479-481: I don't get the point of this sentence, there is no conclusion. Does the fact that the global $N_2O$ budget of Tian et al. (2020) is only based on five global ocean-biogeochemical, weakens the estimates of Tian et al. (2020) ? Or does this fact mean that the $N_2O$ budget of Tian et al. (2020) means that this budget has high uncertainties ?

Line 487: Replace "in that regions" by "in these regions".

Line 489: The authors state that "regional $N_2O$ fluxes by 24%" compared to REF. However, is this number true for climZVAR only, for climZCST only or is this number an average of the overestimation for both climZVAR and climZCST ?

Line 506: Replace "experiments" by "simulations".

Line 515-518: Please rephrase by "The heat perturbations plus the uncertainty... $N_2O$ production result in three $N_2O$ production trajectories through time..."

Line 521: I think it's Northern Hemisphere with capital letters.

Line 528: Replace "experiments" by "simulations".

Line 531: Replace "In forced ocean simulations" by "In ocean-only simulations".

Line 550: Remove "in comparison to REF".

551-552: This sentence can be shortened. Replace "compared to that of the control run REF" by "compared to REF".

Thank you for considering my input to your research.
Rémy Asselot

---

## Author Response (AR2)

Review 2 of Berthet et al. – How does the phytoplankton-light feedback affect the marine N2O inventory?

The authors have carefully revised their manuscript in response to the reviewer comments, and I believe the manuscript has improved substantially as a result and is ready to be published with a few minor revisions. Below I focus on the responses to my own comments and on the revised manuscript. I just have on major comment concerning the steady-state of the model and if this comment is adequately addressed in the revised manuscript, the manuscript can be published.

Thank you for your positive review. We have addressed your comments below. Please find our detailed response (in blue) to your comments (black text). In our responses, lines numbers are relative to the new manuscript.

Major comments

Looking at the response to my previous review and from the revised manuscript, the authors clearly state that their model is in steady-state. I understand that the spin-up is in steady-state after running for 2000 years but I really question the fact that after 20 years of simulations, the model simulations are in steady state. A good argument to say that their simulations are not in steady-state is Fig. 1. This figure shows an increase of OHC300 over the whole simulations (1999-2018). If the model would be in steady-state during the "simulations period", the OHC300 would be constant and not fluctuate anymore after a certain period of time. I think the simulations are too short (only 20 years) to reach a steady state. This point should be clearly stated in the manuscript.

The protocol followed in this study is the classical approach used for all climate hindcasts: first a long spin-up is run to allow the system to reach a steady-state (until 1957 here, see Fig. R1), and then a "transient climate" simulation is forced by historical evolving forcings in order to model the state of the system over the recent period (1958-2018 in this study). The spin-up phase brings the system to an equilibrium in order to avoid intrinsic model drifts during the transient phase.

So we agree that the model is not in steady-state over the period 1958-2018, as this is the transient phase of our simulation. Over this period our ocean-only model is driven by external radiative forcings (JRA55-do) that affect the OHC. Because the global warming is included in these JRA55-do forcings, the positive trend seen in the tropical OHC300 in Fig. 1 is expected over this period. We advocate that the fact that we start from a long spin-up run under constant forcings (through a loop over a specific early period) implies that OHC (and more generally, the ocean) has reached an equilibrium in 1957 (see Fig. R1). Then by applying the observed atmospheric forcing from 1958 onwards, we run a transient climate, and the trend seen in the simulation results primarily from the forcing and does not reflect an intrinsic ocean model drift. The fundamental point is to ensure that we start the transient simulation from a steady ocean in order to not include numerical drifts in the modelled trends after 1958.

Thus it is only during a spin-up period that the ocean heat content (OHC) is expected to reach a stage and stay constant, because by definition a spin-up uses constant forcings (or a loop over a specific early period). By definition, the transient phase is expected to not be in steady-state, and it would be weird to mention it.

Moreover, is PLF taken into account in the spin-up ?

Yes (see Fig. R1), this was stated line 194 of the previous manuscript version: "The control simulation (hereafter REF) together with the spin-up both account for a fully interactive PLF".

[Figure]

Fig. R1: Mean modelled global temperature averaged over the first 300 meters depth. The last spin-up period is shown in gray and REF is in black. The last period of the spin-up (1850-1957) shows that our modelled ocean is in steady-state. From 1958 the complete JRA55-do atmospheric forcings are applied (this is the transient phase), as shown by the clear positive trend which is the signature of the ocean global warming.

Minor comments

The line numbers correspond to the tracked changes version of the manuscript.

Line 34-36: Please rephrase.
Line 41: Replace "effect" by "process".
Line 43: Replace "biophysical" by "biogeophysical" and same for the rest of the manuscript.
Line 44: Replace "which leads" by "leading".
Line 67-69: Replace "is consistent" by "is the same as".
Line 75: Replace "included to" by "included in".
Line 78: Replace "affected" by "affects".
Line 91: Replace "than to the" by "than the".
Line 92-93: Replace "amplifies the mean of PLF-induced changes, but without altering the sign" by "amplifies the magnitude of the PLF-induced changes, without altering the sign".
Line 107-108: is the remote cooling effect at the surface or at the subsurface ?
At the surface.

Line 122: 265-298 times higher ?

Line 257: Replace "consistent" by "identical".

Line 261-262: Please rephrase.

Line 285: Replace "the year" by "when".

Line 287: Replace "OHC of deeper layers" by "OHC in the deeper layers".

Line 297-298: I am not sure by these two lines. Are you results detailed following two different time periods ? One time period covers 10 years (2009-2018) and a second one covers a longer period (1999-2018) ?

Yes time series (Fig. 1, Fig. S3, Fig.4, Fig. 6) present analyses over the whole simulated period (1999-2018).

Line 326: Replace "comparable to that of our" by "comparable to our".

Line 329: Replace "accordance" by "agreement".

Line 331: Replace "order" by "about".

Legend Figure 3: I would say that the 60°S-60°N is not the tropical band anymore. I would encompass that you calculate the average over the tropical and mid-latitudes band.

Line 335: Replace "branching of" by "switching on".

Line 338: Replace "weaker profiles" by "smaller/weaker concentrations".

Line 341: Replace "by comparison" by "compared".

Line 351: Replace "spin-ups run" by "spin-up runs".

In this sentence "run" is a verb.

Legend Figure S4: I am confused by the units. The potential density (y-axis) should be in kg/m3 as in the main text (line 360). The OHC300 (x-axis) should be in ZJ and not ZJ/yr.

This is effectively a mistake, thank you for detecting it ! Fig. S4 presents annual density integrated over the vertical, so why it is in kg/m2, but not in kg/m2/yr. Line 360 has been corrected accordingly. In addition units of the annual mean of OHC300 in the legend of Fig. S4 have also been corrected [ZJ].

Line 360-362: The sentence can be reduced and directly state "The opposite trend (a reduced OHC300 compared to REF) is simulated when considering a variable vertical profile of CHL (climZVAR)."

Line 362-365: This sentence can be reduced as well, giving "However Figure 1 highlights that the simulation REF does not amplify one of these two trends, as climZCST and climZVAR surround REF."

Line 370: Replace "Ranges of uncertainty" by "uncertainty ranges".

Line 373-375: Please repharse.

Legend Figure 6: The units of N2O should be µmolN/m2 and the units of OHC300 should be ZJ.

Thank you, the /yr have been removed.

Line 446: Replace "along time" by "through time".

Line 449: Replace "appear a" by "appears as a".

Line 464: Please define Dpn2o as it is the first time it appears in the main text.

Dpn2o is defined line 294.

Line 466: I guess you mean surface oceanic N2O concentrations.

Yes, of course. As we run ocean-only experiments, we have no atmospheric variables (except in the forcing files). But for clarity we added "oceanic".

Line 479-481: I don't get the point of this sentence, there is no conclusion. Does the fact that the global N2O budget of Tian et al. (2020) is only based on five global ocean-biogeochemical, weakens the estimates of Tian et al. (2020) ? Or does this fact mean that the N2O budget of Tian et al. (2020) means that this budget has high uncertainties ?

Yes, the fact that the budget has been estimated from a small number of ocean-biogeochemical models having different PLF representations add an important uncertainty to it. This point is explained lines 479-483: " *As a matter of fact, the oceanic contribution to the recent global $N_2O$ budget by Tian et al. (2020) is based on only five global ocean-biogeochemical models (as still only few models simulate marine $N_2O$ emissions). These models have different configurations of the PLF which adds considerable uncertainty to simulated marine $N_2O$ emissions*."

Line 487: Replace "in that regions" by "in these regions".
Line 489: The authors state that "regional N2O fluxes by 24%" compared to REF. However, is this number true for climZVAR only, for climZCST only or is this number an average of the overestimation for both climZVAR and climZCST ?

No, it is not an average, the overestimation in each experiment (climZVAR and climZCST) reach 24% in the subtropical gyres of South Pacific and South Atlantic.

Line 506: Replace "experiments" by "simulations".
Line 515-518: Please rephrase by "The heat perturbations plus the uncertainty… N2O production result in three N2O production trajectories through time…"
Line 521: I think it's Northern Hemisphere with capital letters.
Line 528: Replace "experiments" by "simulations".
Line 531: Replace "In forced ocean simulations" by "In ocean-only simulations".
Line 550: Remove "in comparison to REF".
551-552: This sentence can be shortened. Replace "compared to that of the control run REF" by "compared to REF".

Thank you for considering my input to your research.
Rémy Asselot